# Dynamics Reveals Structure: Challenging the Linear Propagation Assumption

Hoyeon Chang [1]    Bálint Mucsányi [2]    Seong Joon Oh [1]

## Abstract

Neural networks adapt through first-order parameter updates, yet it remains unclear whether such updates preserve logical coherence. We investigate the geometric limits of the Linear Propagation Assumption (LPA), the premise that local updates coherently propagate to logical consequences. To formalize this, we adopt relation algebra and study three core operations on relations: negation flips truth values, converse swaps argument order, and composition chains relations. For negation and converse, we prove that guaranteeing direction-agnostic first-order propagation necessitates a tensor factorization separating entity-pair context from relation content. However, for composition, we identify a fundamental obstruction. We show that composition reduces to conjunction, and prove that any conjunction well-defined on linear features must be bilinear. Since bilinearity is incompatible with negation, this forces the feature map to collapse. These results suggest that failures in knowledge editing, the reversal curse, and multi-hop reasoning may stem from common structural limitations inherent to the LPA.

## 1. Introduction

Modern machine learning systems evolve through a long trajectory, spanning pretraining, continual learning (Wu et al., 2024), and unlearning (Jang et al., 2023). Throughout this lifecycle, the central operation for adapting to new information is the first-order parameter update. Ideally, this adaptation should be rational: when a model revises its belief about a fact, its logically related beliefs should update accordingly to maintain coherence. However, maximizing the likelihood of a target fact is fundamentally an optimization process, distinct from rational belief revision (Hase et al., 2024).

[1]KAIST [2]University of Tübingen. Correspondence to: Hoyeon Chang <retapurayo@kaist.ac.kr>, Seong Joon Oh <coallaoh@gmail.com>.

*Proceedings of the 43rd International Conference on Machine Learning*, Seoul, South Korea. PMLR 306, 2026. Copyright 2026 by the author(s).

Consequently, it remains unclear whether the local geometry of gradient-based updates can inherently guarantee such logical consistency without inducing contradictions.

Encouraged by the impressive capabilities of current Large Language Models (LLMs) in logical reasoning during inference (Kojima et al., 2022; Achiam et al., 2023), it is tempting to assume that such coherence is preserved under local, first-order parameter updates. This premise, which we term the **Linear Propagation Assumption (LPA)**, often serves as a foundational design principle in current techniques. For instance, prominent knowledge editing methods explicitly formulate the update as a constrained linear least-squares problem, treating network layers as linear associative memories (Bau et al., 2020; Meng et al., 2022a;b). Beyond editing, this implicit assumption also appears in continual learning strategies that aim to add tasks without forgetting (Lopez-Paz & Ranzato, 2017) and unlearning techniques designed to erase specific knowledge (Jang et al., 2023).

However, the validity of the LPA is questionable, as empirical failures of LLMs on logical coherence show. For instance, LLMs exhibit the "reversal curse," failing to generalize to reverse relationship (Berglund et al., 2023), and struggle with compositional reasoning tasks (Dziri et al., 2023). Since these representations are constructed through first-order updates, such persistent failures suggest that the update mechanism may not reliably imprint the necessary logical structure. This limitation becomes explicitly visible in knowledge editing, where even carefully targeted updates consistently fail to propagate to logical consequences such as negations or implications (Zhong et al., 2023; Cohen et al., 2024; Liu et al., 2025). These phenomena suggest a shared structural issue: the geometry of first-order updates imposes structural constraints that are inherently ill-suited for systematic logical operations.

In this work, we rigorously investigate this hypothesis by asking: **What structural constraints are imposed on a model's representation if we demand that local linear updates respect the logical structure of relational knowledge?** To answer this, we formalize relational knowledge using relation algebra (Givant, 2006), which builds relational knowledge via three fundamental operations: **negation** (flipping truth values), **converse** (swapping argument order), and **composition** (chaining relations). Following Tarski's

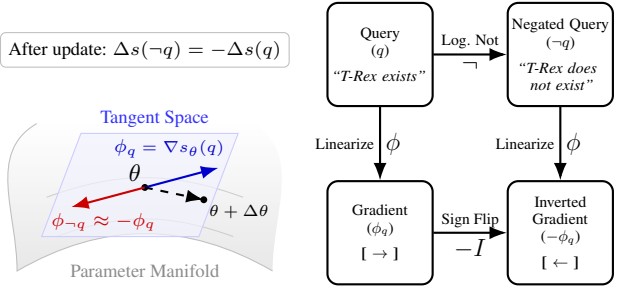

Figure 1. **Geometric interpretation of logical equivariance.** **(Left)** A query $q$ is associated with a score $s_\theta(q)$ (e.g., log probability), and its gradient $\phi_q = \nabla s_\theta(q)$ as a feature. Under the LPA, a local parameter update $\Delta\theta$ induces a score change approximated by the inner product: $\Delta s(q) = s_{\theta+\Delta\theta}(q) - s_\theta(q) \approx \langle \phi_q, \Delta\theta \rangle$. Logical consistency under direction-agnostic first-order propagation requires that any local parameter change enhancing $q$ suppresses $\neg q$ (i.e., $\Delta s(\neg q) = -\Delta s(q)$), which necessitates that the gradient vectors be anti-aligned ($\phi_{\neg q} \approx -\phi_q$). **(Right)** This geometric requirement induces a commutative diagram where symbolic negation $\neg$ in the query space corresponds to a linear inversion $-I$ in the gradient feature space.

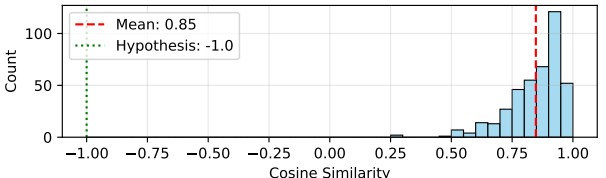

Figure 2. **Gradient alignment hinders negation consistency.** Cosine similarities between gradients of facts and their negations. Contrary to the theoretical requirement for anti-alignment ($= -1$), empirical gradients are strongly positively aligned ($\approx 0.85$). The gradients are computed with respect to the parameters of the last Transformer block and LM head. See App. A for detailed setup.

invariance criterion for logical notions (Tarski, 1986; Sher, 2008), we treat entity renamings as symmetries of the query space and ask what constraints such symmetries impose on the linearized geometry. This provides a principled way to test whether first-order parameter updates can support systematic propagation of logical operations.

Geometrically, satisfying these criteria requires navigating a fundamental trade-off between total superposition (uncontrollable interference) and perfect decoupling (lookup tables). Instead, we seek a *structured coupling*: for example, an update to $p$ must automatically adjust $\neg p$ (Fig. 1), while remaining linearly independent from unrelated facts. We translate this requirement into the geometric structure of gradients to formalize **Systematic Linear Propagation (SLP)**. SLP imposes strict logical equivariance on coupled facts, ensuring that linear updates systematically track unary logical transformations (negation and converse).

Our theoretical analysis reveals several necessary conditions on the required geometry of first-order updates. For negation, we prove that guaranteeing direction-agnostic first-order propagation necessitates a tensor factorization separating entity-pair context from relation content (Theorem 1), reminiscent of Smolensky (1990). Moreover, converse requires its further decomposition into symmetric and antisymmetric components, constraining how argument order is represented (Theorem 2). In contrast, we identify a fundamental obstruction when extending this systematicity to composition (Theorem 3). We show that a minimal form of systematic composition reduces to conjunction, where a conjunction well-defined on linear features necessitates a bilinear structure. However, we demonstrate that this bilinearity is incompatible with the geometry enforced by

negation equivariance. This conflict forces the feature map to collapse, suggesting that the failure to propagate updates to compositional consequences in the first-order regime may stem from a fundamental geometric mismatch.

More broadly, our results support the view that *dynamics reveals structure*: analyzing how representations transform under updates exposes structural necessities that are invisible to static function approximation. This perspective contributes to the classical systematicity debate (Fodor & Pylyshyn, 1988) by deriving binding-compatible block structure not as an architectural choice, but as a geometric necessity for logical coherence under LPA. Ultimately, this opens a path toward **logical geometric deep learning** (Bronstein et al., 2021), treating logical operations not merely as static rules, but as dynamic symmetries to be preserved throughout learning.

## 2. Problem Formulation

This section formalizes the problem. We first show empirically that current LLMs violate a basic requirement for negation consistency under LPA (Sec. 2.1), then introduce relation algebra as our formalism (Sec. 2.2) and linearized features as our geometric tool (Sec. 2.3). These lead to our central definition: Systematic Linear Propagation (SLP) (Sec. 2.4), which grounds the theorems in Sec. 3.

### 2.1. Motivation: Gradient Misalignment in LLMs

The Linear Propagation Assumption (LPA) posits that logical consistency can be maintained via local, first-order parameter updates. For such updates to be rational, the model's geometry must support the logical relations between facts. Specifically, as illustrated in Fig. 1, increasing the score of a query $q$ should automatically suppress that of the negated query $\neg q$. Geometrically, this implies that the update vector $\Delta\theta$ enhancing $q$ should naturally have a negative projection onto the gradient of $\neg q$. In other words, the gradients $\nabla_\theta(q)$ and $\nabla_\theta(\neg q)$ should be **anti-aligned**.

We empirically test this condition on Qwen3-4B and 30B (Yang et al., 2025), and Olmo3-7B (Olmo et al., 2025) using the TREX subset of NEGATED LAMA (Kassner & Schütze, 2020). Contrary to the ideal, the results in Figs. 2 and 4 reveal a misalignment: the gradients of facts and their negations are strongly positively aligned. This indicates that a basic geometric premise for negation-consistent linear score propagation is violated in current LLMs. Consequently, a typical local update improving $q$ will often induce a similar increase in $\neg q$, creating an obstruction for LPA-based methods. We hypothesize this gradient misalignment not merely as a training artifact, but as a symptom of a deeper structural mismatch. Standard representations, shaped by first-order dynamics, seemingly lack the geometric capacity to distinguish a fact from its negation in the tangent space. This motivates our central question: **What representation geometry is required to guarantee systematic logical propagation?**

## 2.2. Preliminary: Relation Algebra

To study the question, we first formalize the logical structure of relational knowledge using relation algebra. The NLP community has predominantly formalized factual knowledge as a collection of relational triples $(h, r, t)$, where a head entity $h$ and a tail entity $t$ are connected by a binary relation $r$ (e.g., (T-Rex, HasA, FourLegs)) (Petroni et al., 2019). This has become a standard protocol for evaluating factual knowledge in LLMs (Cohen et al., 2024), interpreting LLM as an implicit knowledge graph. To analyze this setup mathematically, we adopt the formalism of relation algebra (Givant, 2006). Relation algebra provides a rigorous language for manipulating binary relations, which we use to organize the logical operations we study over relational knowledge modeled as a triplet.

Let $E$ be a finite set of entities, and write $U := E \times E$ for the universe of ordered pairs. A relation on $E$ is a subset $r \subseteq U$. Intuitively, $(h, t) \in r$ means that $h$ stands in relation $r$ to $t$.[1] Now, we write the universe of relations Rel $:= 2^U$ for the set of all binary relations on $E$. Relation algebra consists of Boolean operations alongside relation-specific operations such as converse and composition. In this work, we study two regimes: we first focus on the unary operations negation ($\neg$) and converse ($(\cdot)^{\smile}$), and later turn to the binary operation of composition ($(\cdot; \cdot)$), which underlies multi-hop reasoning. For any relations $r, s \in$ Rel, these operations are defined set-theoretically as:

$$\neg r := U \setminus r, \tag{1}$$
$$r^{\smile} := \{(t, h) : (h, t) \in r\}, \tag{2}$$
$$r; s := \{(h, t) : \exists b \in E, (h, b) \in r \wedge (b, t) \in s\}. \tag{3}$$

---

[1]Throughout, we conflate a relation name with the set of entity pairs for which it holds in the underlying knowledge base.

We focus on these three operations since recent studies identify them as recurring failure modes of LLMs: negation robustness (Kassner & Schütze, 2020; Liu et al., 2025), reversed queries (the "reversal curse") (Berglund et al., 2023), and multi-hop propagation (Cohen et al., 2024).

For our later analysis, let $\mathcal{R}$ be the closure of a base set of relations $\mathcal{R}_0 \subseteq$ Rel under the two unary operations of negation and converse, i.e., the smallest set containing $\mathcal{R}_0$ and satisfying $r \in \mathcal{R} \Rightarrow \neg r \in \mathcal{R}$ and $r^{\smile} \in \mathcal{R}$. We call $(\mathcal{R}, \neg, (\cdot)^{\smile})$ our *unary relation algebra*.[2] Now, given $h, t \in E$ and $r \in \mathcal{R}$, we write $r(h, t)$ for the atomic formula asserting that $(h, t) \in r$. Negation and converse act on these atomic formulas as $\neg r(h, t)$ and $r^{\smile}(t, h)$, respectively. In later sections, we will study how such operations should be represented and propagated in a linear feature space.

## 2.3. Queries, Scores, and Linearized Features

Having established the symbolic structure of relational knowledge, we now bridge the gap to the continuous geometry of neural networks. Our goal is to analyze how logical operations on symbols map to geometric transformations on features. To this end, we operate in the linearized regime of parameter updates, treating the gradient of a query as its feature vector. Note that this approach aligns with the Neural Tangent Kernel (NTK) perspective (Jacot et al., 2018), effectively treating the model as a linear function of parameters defined by these gradient features.

**Queries.** We model each atomic formula as a triplet of a head entity, a relation, and a tail entity. Let $E$ be a finite set of entities and let $\mathcal{R}$ be our unary relation algebra. A query is a triple $q = (h, r, t) \in Q := E \times \mathcal{R} \times E$, where a query $q = (h, r, t)$ can be read as an atomic formula $r(h, t)$ as discussed above.

**Scores and Linearization.** Fix a finite-dimensional real inner-product space $(\Theta, \langle \cdot, \cdot \rangle)$ representing the parameters of a differentiable model, and let $\theta_0 \in \Theta$ be a reference model state. We associate each query $q \in Q$ with a differentiable scalar score $s_\theta(q) \in \mathbb{R}$ (e.g., the logit of the correct tail entity). We assume that the model's decision about $q$ (e.g., predicted truth or preference) is determined by a rule strictly monotone in $s_\theta(q)$. As we are primarily interested in how these scores change under the first-order regime, we define the feature of a query as follows:

**Definition 1** (Linearized feature). *For each query $q \in Q$, we define its* linearized feature *at $\theta_0$ as the gradient of the score:*

$$\phi_q := \nabla_\theta s_\theta(q)\big|_{\theta = \theta_0} \in \Theta.$$

*Consequently, the score change under a local parameter*

---

[2]We later return to consider composition in Sec. 4.

*update $\Delta\theta$ is given by the first-order Taylor approximation:*

$$s_{\theta_0 + \Delta\theta}(q) \approx s_{\theta_0}(q) + \langle \phi_q, \Delta\theta \rangle.$$

The features $\phi_q$ typically span only a subspace of the full parameter space. Henceforth, we restrict our analysis to the subspace $W := \text{span}\{\phi_q : q \in Q\} \subseteq \Theta$, since components of an update $\Delta\theta$ orthogonal to $W$ do not affect first-order score changes on the queries we study. We empirically validate the practical relevance of this first-order approximation for local LLM updates in Sec. A.2.

### 2.4. Systematic Linear Propagation

We now unify the algebraic structure of knowledge (Sec. 2.2) with the geometric view of features (Sec. 2.3). Our goal is to formalize when a linearized model supports the *systematic* propagation with respect to the unary relation algebra we consider. Qualitatively, systematicity imposes structured coupling in the feature space: an update to a query $q$ should propagate to its converse and inversely to its negation, while logically unrelated facts remain independently controllable.

A key choice in our formulation is that we do *not* define systematicity as the mere existence of a carefully selected update direction that satisfies these constraints. Instead, we ask for **automaticity**: logical coupling must be an intrinsic geometric property of the representation. This ensures that once an update is applied to enforce $q$, the induced effects on logically related queries follow *automatically*, without requiring the optimizer to solve a separate constraint-satisfaction problem. Formally, we require the logical constraints to hold for all $\Delta\theta \in W$ in the first-order regime.

This direction-agnostic requirement is motivated by the potential fragility of relying on specific "safe" update directions in high-capacity models. In practical regimes where the feature space is highly superposed (Elhage et al., 2022; Hu et al., 2025), reliably identifying an update direction that satisfies logical constraints while avoiding interference with unrelated facts is often prohibitively difficult. Furthermore, in lifelong learning settings, any such transient "safe subspace" itself might be prone to drift as the representation evolves. Thus, by enforcing systematicity for all update directions, we isolate a notion in which logic is an intrinsic property of the local geometry rather than an artifact of a specific optimization trajectory or a transient subspace.

To formalize this, we first identify the closed sets of queries that must be logically coupled under any update. Recall that our unary relation algebra $\mathcal{R}$ is closed under negation $\neg$ and converse $(\cdot)^{\smile}$. These relational operations induce a corresponding action on the space of queries $Q$ as follows:

$$\neg(h, r, t) := (h, \neg r, t), \tag{4}$$
$$\text{rev}(h, r, t) := (t, r^{\smile}, h). \tag{5}$$

Intuitively, $\neg$ flips the truth value by acting locally on the relation slot (e.g., $\neg\text{ChildOf} \to \text{NotChildOf}$ where $\text{ChildOf} \subseteq U$ and $\text{NotChildOf} := U \setminus \text{ChildOf}$), while keeping entities fixed. In contrast, rev swaps the head and tail entities while moving to the converse relation (e.g., $\text{ChildOf}^{\smile} \to \text{ParentOf}$), thereby preserving the truth value (i.e., $r(h, t) \Leftrightarrow r^{\smile}(t, h)$).

Observe that both operations are involutions and commute:

$$\neg(\neg q) = q, \quad \text{rev}(\text{rev}(q)) = q, \quad \text{rev}(\neg q) = \neg(\text{rev}(q)).$$

Hence, these two logical operations generate a group $G := \{\text{id}, \neg, \text{rev}, \neg \circ \text{rev}\}$ acting on $Q$. We define the orbit of a query $q$ under this group as its **logical family**:

$$G \cdot q := \{g(q) : g \in G\}.$$

In plain words, two queries lie in the same family iff one can be transformed into the other via the group operations.

We now translate the semantic requirements of these families into constraints on the feature space $W$. As discussed above, we require that the score changes are coordinated for *all* update directions $\Delta\theta$. For negation, the condition $\Delta s(\neg q) = -\Delta s(q)$ implies that the feature vectors are strictly anti-aligned: $\phi_{\neg q} = -\phi_q$. Applying the same logic to the converse operation yields $\phi_{\text{rev}(q)} = \phi_q$. Therefore, we formalize this requirement as follows:

**Definition 2** (Logical equivariance). *A feature map $\phi : Q \to W$ is* logically equivariant *w.r.t. $\neg$ and* rev *if*

$$\phi_{\neg q} = -\phi_q, \qquad \phi_{\text{rev}(q)} = \phi_q \quad \text{for all } q \in Q.$$

While Def. 2 ensures coupling *within* families, we must also ensure that logically unrelated facts do not interfere with each other. If feature vectors from disjoint logical families are linearly dependent, it becomes impossible to modify one family without inadvertently affecting another. Therefore, to enable selective update, distinct logical families must be linearly independent in the feature space. We formulate these two requirements, intra-family coupling and inter-family decoupling, as **Systematic Linear Propagation (SLP)**.

**Definition 3** (Systematic Linear Propagation (SLP)). *A linearized model $\{\phi_q\}_{q \in Q}$ is said to satisfy* Systematic Linear Propagation *with respect to the unary relation algebra if:*

1. *It is logically equivariant in the sense of Def. 2*

2. *There exists a choice of one representative query from each family s.t. the corresponding feature vectors are linearly independent in $W$.*

We emphasize that SLP is a deliberately strong formalization of the informal Linear Propagation Assumption: since

only the projection of an update onto $W$ affects first-order score changes, we require the logical constraints to hold for all $\Delta\theta \in W$. Thus, failing SLP does not imply that propagation is impossible, but rather that it cannot be guaranteed by direction-agnostic first-order geometry alone and must rely on additional structure (e.g., restricted update directions, nonlinearity, or higher-order effects). In the following section, we derive the geometric structure theoretically required for a model to support SLP.

## 3. SLP Induces Tensor-Factorized Features

Having formalized SLP, we derive the structural implications of these constraints. In Sec. 3.1, we prove that demanding systematic propagation with respect to negation requires the decomposition of feature space into a tensor product structure separating *context* (i.e., entity pair) from *relation*. Next, we prove in Sec. 3.2 that requiring systematic propagation with respect to converse further decomposes the entity pair component into symmetric and antisymmetric parts, encoding the directionality of relations.

### 3.1. Negation Equivariance Forces Tensor Factorization

To rigorously derive the geometric structure, we adopt Tarski's criterion (Tarski, 1986; Sher, 2008) as our guiding principle. It posits that logical notions are characterized by their invariance under all permutations of the domain's objects. In our context, this implies that logical operations must be invariant to the specific identities of the entities involved. For instance, the logic of negation should apply equally to `T-Rex` and `Chicken`. We formalize this requirement by identifying the symmetry group acting on the query space $Q$. Let $G_E := \mathrm{Sym}(E)$ be the permutation group acting on the set of entities $E$. Following Tarski's criterion, the system must remain consistent under any entity renaming $\sigma \in G_E$, which acts on queries by renaming entities uniformly: $\sigma \cdot (h, r, t) := (\sigma(h), r, \sigma(t))$. Simultaneously, the negation operation ($\neg$) is an involution ($\neg(\neg q) = q$), generating the cyclic group of order 2, denoted as $\mathbb{Z}_2 = \{1, -1\}$.

Crucially, **entity renaming commutes with logical operations:** renaming entities does not alter the logical relationship, and logical transformations do not affect entity identities, i.e., $\sigma \cdot (\neg q) = \neg(\sigma \cdot q)$. Consequently, we can define a product group $H := G_E \times \mathbb{Z}_2$, which acts on the set of queries $Q$ via the simultaneous action of renaming and optional negation. We translate these symbolic symmetries into the geometry of the feature space $W$. As formally verified in Lemma 7, under SLP, it is guaranteed that the action of $H$ on the query space induces a well-defined lin-

ear group representation of $H$ on the feature space $W$.[3][4] Since $W$ is a finite-dimensional representation of the product group $H = G_E \times \mathbb{Z}_2$, we can utilize standard results in representation theory to prove the following factorization theorem.

**Theorem 1** (Context-Relation Factorization (Proof in App. C)). *Let $\phi : Q \to W$ be the feature map defined by $q \mapsto \phi_q$. If $\phi$ satisfies SLP, then there exist real vector spaces $\{C_i\}_i, \{R_i\}_i$ and an isomorphism $W \cong \bigoplus_i (C_i \otimes R_i)$ such that*

$$\phi(h, r, t) \;=\; \bigoplus_i \left( \sum_{k=1}^{m_i} u_{i,k}(h,t) \otimes v_{i,k}(r) \right),$$

*where $u_{i,k} : E \times E \to C_i$ and $v_{i,k} : \mathcal{R} \to R_i$. Moreover, negation acts locally as a sign flip on each relation component, i.e., $v_{i,k}(\neg r) = -v_{i,k}(r)$.[5]*

Theorem 1 indicates that, to support systematic linear propagation with respect to negation, the feature geometry must separate entity-pair context information from relation information. Specifically, logical negation is realized via local negation on the relation factors $v_{i,k}(r)$. In Sec. 6, we discuss how this slot-local action relates to the problem of variable binding (Greff et al., 2020).

### 3.2. Converse Equivariance Forces Positional Alignment

Having established that logical equivariance under negation forces a blockwise tensor factorization, we now investigate the structural implications of the converse operation. Recall that the symbolic converse operation swaps the head and tail entities and replaces $r$ with its converse $r^{\smile}$, i.e., $\mathrm{rev}(h, r, t) = (t, r^{\smile}, h)$. SLP requires the feature map to be invariant under this operation, i.e., $\phi_{\mathrm{rev}(q)} = \phi_q$. This constraint forces a parity alignment between the context and relation components, as stated in the following theorem.

**Theorem 2** (Symmetric-Antisymmetric Alignment (Proof in App. D)). *Assume the context-relation factorization from Theorem 1 and suppose $\phi$ is* converse-invariant, *i.e.*

$$\phi(h, r, t) = \phi(t, r^{\smile}, h) \qquad \text{for all } (h, r, t) \in Q.$$

*Then for each block $i$, the corresponding feature component $\phi_i$ admits a decomposition with matched-parity:*

$$\phi_i(h, r, t) = \phi_i^+(h, r, t) + \phi_i^-(h, r, t),$$

---

[3]For readers unfamiliar with these concepts, we provide a brief primer on group representation theory in App. B.

[4]In this section, we use 'representation' exclusively in the sense of group representation theory, whereas we refer to the ML concept of embeddings strictly as 'features' to prevent ambiguity.

[5]We use $\bigoplus_i W_i$ for a direct-sum decomposition, meaning that a feature vector decomposes into separate components across the blocks $W_i$. We use $C \otimes R$ for a tensor product, which formalizes binding a context (entity-pair) factor in $C$ with a relation factor in $R$.

*where $\phi_i^{\pm}$ can be chosen as a sum of context-relation tensor terms $u(h, t) \otimes v(r)$ satisfying*

$$u(t, h) = \pm u(h, t) \quad and \quad v(r^{\smile}) = \pm v(r).$$

This result reveals a geometric account of how directionality can be represented under converse invariance. The first term represents symmetric components, where neither the relation nor the entity pair cares about order. In contrast, the second term $u^- \otimes v^-$ encodes directionality through a mechanism of **sign cancellation**: swapping the entities introduces a sign flip $(-1)$ in the context factor $u(h, t)$, which is exactly compensated by the sign flip in the relation factor $v(r)$ induced by the converse operation.

Note that the constraints from negation and converse are compatible, as the corresponding symbolic operations commute. In conclusion, while SLP is theoretically realizable through this specific symmetry, such a structure is likely absent in the unconstrained feature spaces of practical LLMs, as demonstrated in Sec. 2.1.

## 4. The Collapse of Linear Conjunction

In this section, we analyze systematic propagation for relational composition (i.e., multi-hop reasoning). We first show that systematic relational composition must handle conjunction in a minimal subclass. We then identify an obstruction to conjunction under our linearized systematicity requirement. In particular, if conjunction must be well-defined on the linearized feature space (Assumption 1), then conjunction-faithful propagation is incompatible with negation equivariance, unless the feature map is a zero map.

We seek systematic propagation to compositional consequences: a targeted update to an atomic formula on a relation $r \in \mathcal{R}$ must automatically adjust composed queries involving a composition $r; s$ for some $s \in \mathcal{R}$. That is, by systematicity, we require that the feature of $r; s$ be modeled *constructively* from its constituents to ensure propagation under arbitrary parameter updates that change $r$. Recall from Eq. (3) that relational composition, $(r; s)(h, t) \iff \exists b : r(h, b) \wedge s(b, t)$, links two queries via an intermediate entity. While the general case involves existential aggregation over possibly many witnesses, we isolate a minimal subclass that removes aggregation altogether. Specifically, in the *unique-witness* case where there exists a unique $b^*$ satisfying the link, composition reduces to the conjunction $r(h, b^*) \wedge s(b^*, t)$. Thus, any mechanism that supports systematic composition must, at a minimum, support systematic conjunction. We therefore investigate whether conjunction can be realized in the linearized feature space while remaining compatible with negation equivariance.

We first characterize the constraints LPA imposes on feature geometry to ensure that conjunction is supported system-

atically. Central to this relationship is compositionality: the representation of a compound statement should be systematically determined by its constituents. That is, given a compound query $p \wedge q$ with $p, q \in Q$, its feature $\phi_{p \wedge q}$ should be determined solely by $\phi_p$ and $\phi_q$. Along with the logical properties of sentential conjunction (commutativity and idempotence), we formalize this intuition as follows:

**Definition 4** (Conjunction-faithful features under LPA). *Let $Q^{\wedge}$ be the closure of $Q$ under negation and conjunction, i.e., the smallest set of queries containing $Q$ and closed under both operations. Let $W^{\wedge} := \mathrm{span}\{\phi_p : p \in Q^{\wedge}\}$. We say that $\phi$ is conjunction-faithful (under LPA) if there exists a binary operator $F : W^{\wedge} \times W^{\wedge} \to W^{\wedge}$ that governs the conjunction of features, satisfying the following properties for all $p, q \in Q^{\wedge}$ and $u, v \in W^{\wedge}$:*

*(i) Consistency: $\phi_{p \wedge q} = F(\phi_p, \phi_q)$.*

*(ii) Symmetry: $F(u, v) = F(v, u)$.*

*(iii) Idempotence: $F(u, u) = u$.*

The consistency condition states that the feature of a conjunction depends solely on the features of its conjuncts. Consequently, two formulas with identical features must behave identically under conjunction with any fixed context:

**Lemma 1** (Substitution). *If $\phi$ is conjunction-faithful, then for all $p, p', q \in Q^{\wedge}$, $\phi_p = \phi_{p'}$ implies $\phi_{p \wedge q} = F(\phi_p, \phi_q) = F(\phi_{p'}, \phi_q) = \phi_{p' \wedge q}$.*

While Lemma 1 guarantees that conjunction is a well-defined function of features, the linearity of LPA imposes a stronger constraint. Under the first-order regime, the editing dynamics are governed entirely by linear projections $(\Delta s(p) = \langle \phi_p, \Delta\theta \rangle)$. This implies that the editor cannot distinguish any linear dependence: if a weighted sum of features is zero $(\sum_i a_i \phi_{p_i} = 0)$, the collective score change is identically zero for any parameter update.

Geometrically, such a zero-sum combination constitutes information that is operationally non-existent to the model's update mechanism. If a conjunction operator were to map a null signal to a non-zero feature, it would generate distinctions based on information invisible to the editor, thereby decoupling the logic of propagation from the physics of editing. To ensure that propagation remains predictable from first-order geometry alone, we adopt a strong notion of systematicity: logical operations must be consistent with the linear geometry of the editor, meaning they must preserve linear dependencies. Formally, this requires that if a linear combination vanishes $(\sum_i a_i \phi_{p_i} = 0)$, its conjunction with any context $q$ must also vanish $(\sum_i a_i \phi_{p_i \wedge q} = 0)$. This condition is formalized as the following assumption:

**Assumption 1** (Kernel stability for conjunction). *Let $V^{\wedge} := \mathrm{span}\{e_p : p \in Q^{\wedge}\}$ be the free real vector space on $Q^{\wedge}$, and let $\Phi : V^{\wedge} \to W^{\wedge}$ be the linear extension defined by*

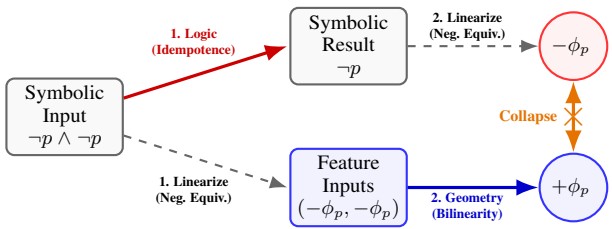

*Figure 3.* **The incompatibility of logical conjunction and LPA.**
**Top Path:** Logical idempotence maps $(\neg p, \neg p) \to \neg p$, expecting
feature $-\phi_p$. **Bottom Path:** Linearization gives $(-\phi_p, -\phi_p)$,
and the bilinearity of $\tilde{F}$ yields $+\phi_p$. The only possible way to
commute the two paths is setting $\phi_p = 0$, leading to a collapse.

$\Phi(e_p) := \phi_p$. *For each fixed $q \in Q^\wedge$, define the linear map
$T_q : V^\wedge \to V^\wedge$ by $T_q(e_p) := e_{p \wedge q}$. Assume that*

$$T_q(\ker \Phi) \subseteq \ker \Phi \qquad \text{for all } q \in Q^\wedge.$$

Mathematically, this condition ensures that the conjunction operation is well-defined on $W^\wedge$. If Assumption 1 fails, then conjunction cannot be systematically treated as a well-defined operation compatible with the linearized feature geometry, as it may distinguish linear combinations of formulas whose difference is invisible to all first-order updates. In that case, it becomes difficult to maintain the premise that compositional reasoning can emerge through linear propagation alone.

Under this assumption, we can prove that a feature-level conjunction of two queries is fully characterized by a bilinear operator.

**Lemma 2** (Kernel Stability Yields Bilinearity (Proof in App. E))**.** *Assume Assumption 1 and the symmetry of conjunction (Def. 4(ii)). Then, there exists a unique symmetric bilinear operator $\tilde{F} : W^\wedge \times W^\wedge \to W^\wedge$ such that $\tilde{F}(\phi_p, \phi_q) = \phi_{p \wedge q}$ for all $p, q \in Q^\wedge$.*

Lemma 2 implies that once conjunction is required to behave systematically under LPA, its behavior on features is completely governed by the induced bilinear operator $\tilde{F}$. However, the theorem below demonstrates that no non-trivial bilinear feature map can satisfy idempotence.

**Theorem 3** (Structural Collapse of Linear Conjunction)**.** *Let $\tilde{F}$ be the bilinear map from Lemma 2. If $\phi$ satisfies idempotence (Def. 4) and negation equivariance ($\phi_{\neg p} = -\phi_p$), then $\phi_q = 0$ for all $q \in Q^\wedge$.*

*Proof.* Let $p \in Q$ be an atomic query and let $u = \phi_p$. From Idempotence, we have $\tilde{F}(u, u) = \phi_{p \wedge p} = \phi_p = u$. By negation equivariance, $\phi_{\neg p} = -u$. Since $\neg p \in Q^\wedge$, we may apply idempotence to $\neg p$ as well:

$$\tilde{F}(-u, -u) = \tilde{F}(\phi_{\neg p}, \phi_{\neg p}) = \phi_{\neg p \wedge \neg p} = \phi_{\neg p} = -u.$$

On the other hand, since $\tilde{F}$ is bilinear,

$$\tilde{F}(-u, -u) = (-1)(-1)\tilde{F}(u, u) = \tilde{F}(u, u) = u.$$

Comparing the two results, we have $-u = u$, which implies $u = 0$. Thus, $\phi_p = 0$ for all atomic queries. Since any compound query $q \in Q^\wedge$ is formed by finite conjunctions of atomic queries (i.e., $q = p_1 \wedge \cdots \wedge p_k$ where $p_i \in Q$) and $\tilde{F}(0, \cdot) = 0$, by induction, $\phi_q = 0$ for all $q \in Q^\wedge$. $\square$

Intuitively, the collapse stems from a fundamental conflict between the symbolic rules of logic and the algebraic rules of bilinearity, as illustrated in Fig. 3. Specifically, the diagram illustrating the correspondence between logic and geometry fails to commute. Following the logical path, the rules of negation and idempotence dictate that the result must flip sign ($u \to -u$), preserving the negation. In contrast, following the geometric path, the bilinearity of $\tilde{F}$ forces the signs to cancel out ($(-u, -u) \to u$), as the product of two negatives is positive. The only vector satisfying both requirements is the zero vector, leading to the collapse.

One might wonder whether this obstruction is merely an artifact of exact negation equivariance. However, as shown in App. F, the conflict persists even when negation is only approximately represented as a sign flip, unless the induced bilinear conjunction becomes increasingly ill-conditioned. Thus, the structural collapse of linear conjunction does not vanish gracefully under small relaxations of exact logical equivalence.

This result implies that, in the first-order regime, the geometries required for negation and conjunction are incompatible. Under our systematicity conditions that render conjunction as a bilinear operation on features, enforcing negation equivariance collapses the representation. Importantly, this obstruction is structural rather than an optimization failure. It therefore casts doubt on the feasibility of systematic multi-hop propagation under local linear updates, consistent with empirical breakdowns, as discussed in Sec. 6.

## 5. Related Work

**Linearized Dynamics and Model Adaptation.** A common lens in modern deep learning is to approximate adaptation via local updates in a linearized feature space, a perspective theoretically grounded in Neural Tangent Kernel (NTK) analyses (Jacot et al., 2018; Lee et al., 2019). This viewpoint arises across settings that rely on gradient-based updates, including pretraining trajectories that accumulate factual associations (Chang et al., 2024), as well as domain adaptation and continual learning where updates interact with prior knowledge (Gururangan et al., 2020; Wu et al., 2024). Most explicitly, knowledge editing methods such as ROME and MEMIT treat Transformer MLPs as key-value memories (Geva et al., 2021), implementing edits as constrained

least-squares updates (Meng et al., 2022a;b). Related first-order schemes also appear in unlearning approaches aimed at reducing the influence of specific data (Jang et al., 2023; Barez et al., 2025). A recurring empirical theme is that local updates do not reliably generalize to logically related or compositional variants of the target behavior (Cohen et al., 2024; Liu et al., 2025), echoing broader concerns about the gap between optimization and belief revision (Hase et al., 2024).

**Systematicity and Variable Binding.** The systematicity debate argues that connectionist models may lack the structural machinery for compositional generalization and variable binding (Fodor & Pylyshyn, 1988; Lake & Baroni, 2018). While Smolensky (1990) proposed Tensor Product Representations as a mechanism for variable binding, the binding problem remains an active challenge in modern deep learning (Greff et al., 2020). Recent interpretability work studies linear representations in neural activations (Park et al., 2023), yet empirical studies report persistent failures of compositional generalization in LLMs (Dziri et al., 2023; Wang et al., 2024a; Chang et al., 2025). In particular, Wang & Sun (2025) hypothesized that failures of variable binding underlie the reversal curse. These threads motivate analyzing what structures are required for systematic behavior.

**Logical Structure as Geometric Invariance.** To formalize logical structure in vector spaces, we adopt invariance-based views of logical notions (Tarski, 1986; Sher, 2008). This aligns with geometric deep learning, which characterizes representations by invariance and equivariance (Bronstein et al., 2021; Cohen & Welling, 2016). This perspective motivates treating logical operations as transformations on queries and studies the constraints they induce in a linearized feature geometry.

## 6. Discussion

Our investigation establishes a bridge between the algebraic structure of logic and the linear geometry of knowledge representations. **A central insight of our work is that the structure of knowledge is best observed through its *dynamics*, i.e., how representations coordinate under first-order updates.** While expressive networks may statically realize a logically coherent representation, our results (Sec. 3, Sec. 4) show that preserving such coherence under linearized updates imposes strict geometric constraints that are invisible to static analyses. This distinction between expressivity and systematic propagation provides a geometric explanation for when logical structure can be realized and when it can be preserved under first-order updates.

**Implications for Systematicity.** As discussed in Sec. 5, tensor products were proposed as a sufficient mechanism

for variable binding. Our results sharpen this connection in the linearized update regime: requiring systematic linear propagation forces a blockwise tensor factorization of the representation (Sec. 3). In this sense, our analysis suggests that such a mechanism is not only an architectural choice, but can be a geometric necessity for preserving logical structure under first-order dynamics. Moreover, our converse analysis (Sec. 3.2) shows that systematic propagation constrains how argument order is encoded, requiring sufficient positional structure to remain consistent under reversal. Relatedly, even when a concept is linearly decodable at a fixed parameter state, static linearity alone does not guarantee systematic interaction under local updates.

**Toward Logical Geometric Deep Learning.** Our results align with geometric deep learning by casting systematic propagation as an equivariance constraint of logical operations. This motivates a direction toward *logical* geometric deep learning, where update-time behavior is constrained by these symmetries beyond static function approximation. One way to view this is that learning dynamics should remain near a constrained manifold defined by logical equivariance, instead of drifting into regions where logically related queries become entangled. Architecturally, this points to parameterizations and objectives that enforce the structures required by systematic propagation, for example, by regularizing gradient features or by building equivariant modules that preserve these symmetries across updates.

**Implications for Model Adaptation and Editing.** This geometric perspective has direct implications for practical model adaptation, most notably knowledge editing. Locate-and-edit methods assume that knowledge can be localized and manipulated through a single local update, but our results emphasize that the relevant structure is defined by how representations transform under that update. This shifts attention from *where* a fact is stored to *whether* the representation supports stable, structure-preserving update directions, i.e., *editability* as a geometric property (Sinitsin et al., 2020). At the same time, our conjunction result suggests that approaches relying on local linearity may face fundamental limits when asked to propagate edits to compositional consequences. This points to mechanisms beyond single-step locality, such as iterative nonlinear updates or memory-based updates that bypass single-update bottlenecks (Mitchell et al., 2022; Wang et al., 2024b).

**Scope and Open Questions.** We view our results as a stress test of systematic propagation in a deliberately strict, first-order setting. We impose direction-agnostic equivariance as a strong requirement, yielding sharp geometric constraints and an impossibility result. Our results are local by design: they characterize what first-order geometry can guarantee at a reference state $\theta_0$. For multi-step optimization, a

natural extension is the NTK or lazy-training regime, where tangent features remain approximately constant along training and gradient descent is governed by the same linearized geometry over time (Jacot et al., 2018; Lee et al., 2019; Chizat et al., 2019). In this regime, our obstruction applies stepwise rather than only to a single isolated update. By contrast, for genuinely non-local optimization where higher-order effects substantially change the tangent features, the obstruction in Theorem 3 need not directly apply. Thus, an important direction is to understand which conclusions persist under weaker, approximate notions of equivariance, and how architectures or objectives can encourage the required structure over repeated updates. Taken together, our results motivate the design of "logical" inductive biases, and suggest a concrete program for testing whether enforcing such structure can support systematic propagation.

## 7. Conclusion

In this work, we analyzed the geometric limits of the Linear Propagation Assumption (LPA). We showed that demanding systematic propagation imposes strict structural constraints: negation and converse require a blockwise tensor-product decomposition (Theorems 1 and 2), while conjunction is fundamentally incompatible with negation under linearity (Theorem 3). These constraints suggest that failures of local linear updates are not merely optimization artifacts, but may reflect a mismatch between the algebraic structure of logic and first-order update geometry. More broadly, failures in knowledge editing, the reversal curse, and multi-hop reasoning may share a common geometric origin in the limits of first-order propagation under the LPA. Ultimately, our findings reinforce the view that **dynamics reveals structure**: logical coherence is constrained not only by what a model can represent, but by how it can be updated.

## Impact Statement

This paper presents a theoretical analysis of the geometric limits of first-order model adaptation in language models. Our goal is to clarify what local, approximately linear update mechanisms can and cannot guarantee about preserving logical coherence. By characterizing structural constraints required for systematic propagation under such updates, our findings caution against naive reliance on single-step local interventions in settings where compositional consistency matters. This perspective is relevant not only to knowledge editing, but also to continual learning and unlearning, where updates are expected to modify specific behaviors without introducing hard-to-detect logical inconsistencies. We hope this contributes to safer deployment by motivating update mechanisms and inductive biases that explicitly support structure-preserving adaptation.

## Acknowledgements

The authors thank Jaewon Oh, Hanseul Cho, Jaden Park, and Jinwoo Heo for comments and discussions.

This work was supported by Institute for Information & communications Technology Planning & Evaluation(IITP) grant funded by the Korea government (MSIT) (RS-2019-II190075, Artificial Intelligence Graduate School Program(KAIST)). Hoyeon Chang acknowledges support by Institute of Information & communications Technology Planning & Evaluation (IITP) grant funded by the Korea government (MSIT) (No.RS-2019-II190075 Artificial Intelligence Graduate School Program (KAIST), 10%; No.RS-2021-II212068, Artificial Intelligence Innovation Hub, 10%; RS-2024-00398115, Research on the reliability and coherence of outcomes produced by Generative AI, 20%; No.2022-0-00113, Developing a Sustainable Collaborative Multi-modal Lifelong Learning Framework, 20%; No.RS-2022-II220264, Comprehensive Video Understanding and Generation with Knowledge-based Deep Logic Neural Network, 20%; RS-2024-00397966, Development of a Cybersecurity Specialized RAG-based sLLM Model for Suppressing Gen-AI Malfunctions and Construction of a Publicly Demonstration Platform) and the InnoCORE program of the Ministry of Science and ICT(N10250156).

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

# A. Experimental Details and Additional Results

## A.1. Gradient Alignment Experiment

To empirically investigate whether recent LLMs respect negation equivariance Def. 2, we measure the geometric alignment between the gradients of factual queries and their negated counterparts.

**Models.** We evaluated our hypothesis across models of varying sizes and architectures to ensure the generality of our findings. Specifically, we used the instruction-tuned versions of the Qwen series (Yang et al., 2025): `Qwen3-4B-Instruct-2507` and `Qwen3-30B-A3B-Instruct-2507`. Additionally, to verify cross-architecture consistency, we included `OLMo-3-7B-Instruct` (Olmo et al., 2025).

**Dataset.** We utilized the TREX subset of the NEGATED LAMA benchmark (Kassner & Schütze, 2020). This dataset provides pairs of cloze-style queries: a factual query $p$ (e.g., "The capital of France is [MASK]") and its negated form $\neg p$ (e.g., "The capital of France is not [MASK]"). We sampled 10 instances for each of 41 templates, making 410 datapoints in total.

**Gradient Computation.** We define the score function $s_\theta(q)$ as the log-probability of the correct target span given the prompt. We compute the gradient of the sequence-level log-probability for the entire target entity span. Formally, for a target span $y = (y_1, \ldots, y_L)$ and prompt $x$, the score is:

$$s_\theta(q) = \log P_\theta(y|x) = \sum_{t=1}^{L} \log P_\theta(y_t|x, y_{<t}) \tag{6}$$

We computed the gradients $\phi_p = \nabla_\theta s_\theta(p)$ and $\phi_{\neg p} = \nabla_\theta s_\theta(\neg p)$. To analyze the geometry of the learned representations rather than the entire parameter space, we restricted the gradient computation to the parameters of the last transformer block and the language modeling head. This restriction is motivated by two key factors: (1) computational feasibility, as calculating full-parameter gradients for every sample is prohibitively expensive, and (2) semantic relevance, as high-level semantic concepts are known to be predominantly encoded in the later layers of the network (Meng et al., 2022a).

**Metric.** We measured the cosine similarity between the two gradient vectors:

$$\text{Sim}(p, \neg p) = \frac{\langle \phi_p, \phi_{\neg p} \rangle}{\|\phi_p\| \|\phi_{\neg p}\|} \tag{7}$$

**Implementation Details.** All experiments were conducted using the PyTorch framework on a node equipped with 4 NVIDIA A100 (80GB) GPUs. The total runtime for the full evaluation pipeline across all models and relations was approximately 10 hours.

**Additional Results on Scale and Architecture.** In the main text (Fig. 2), we report that `Qwen3-4B` exhibits a strong positive alignment (Mean: 0.85) between factual and negated gradients, contradicting the theoretical requirement for systematic propagation. Here, we demonstrate that this phenomenon is not specific to a single model size or architecture. Specifically, we extended our analysis to `Qwen3-30B` (Fig. 4a) and `OLMo-3-7B` (Fig. 4b). As shown in the histograms, both models display a similar distribution heavily skewed towards positive cosine similarity. Specifically, for Qwen3-30B, Fig. 4a shows a high mean (0.86) similar to the 4B model, suggesting that simply scaling up the model may not resolve this tendency. Moreover, Fig. 4b shows a slightly lower but still positive mean (0.62). Overall, these results suggest that negation equivariance does not hold in current LLMs, regardless of model scale or architecture.

## A.2. Validity of the Linearized Update Approximation

To support the practical relevance of the first-order regime, we evaluate how well actual model updates are predicted by the linearized score approximation. We use Qwen3-4B-Instruct and sample 100 instances from the TREX subset of NEGATED LAMA. For each instance, following common local editing setups that target MLP layers (Meng et al., 2022a;b), we compute the target-logit gradient $g$ with respect to the parameters of layer 10 and update the parameters along the normalized gradient direction $g/\|g\|$ with step size $\eta$. The first-order prediction for the target-logit change is then $\eta\|g\|$. We compare this prediction with the actual target-logit change after the update by fitting a linear regression across instances.

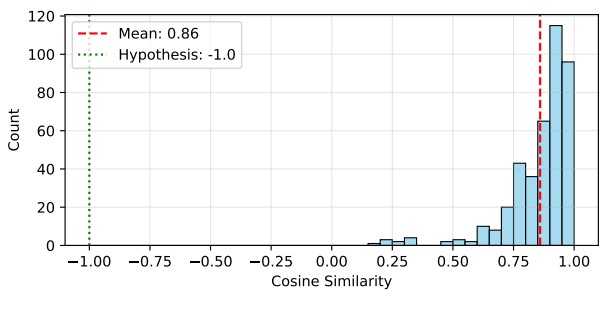
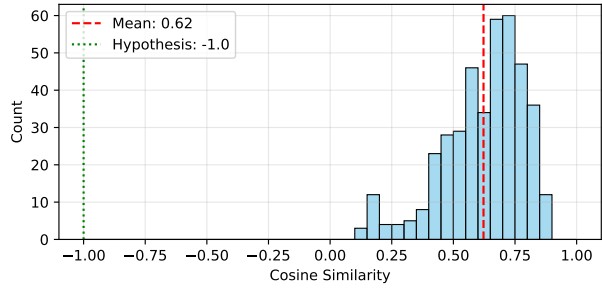

*(a)* **Qwen3-30B**                  *(b)* **OLMo-3-7B**

*Figure 4.* **Gradient alignment across different scales and architectures.** The positive alignment phenomenon persists in (a) Qwen3-30B and (b) OLMo-3-7B. Both distributions are heavily skewed towards positive cosine similarity, demonstrating that the geometric mismatch for linear negation propagation is consistent across model scale and architecture.

*Table 1.* Validity of the linearized update approximation. Actual target-logit changes are well predicted by the first-order approximation for small to moderate step sizes, while the approximation breaks down for very large updates.

| Step size $\eta$ | Slope | $R^2$ |
|---|---|---|
| $10^{-5}$ | 0.97 | $> 0.99$ |
| $10^{-4}$ | 1.00 | $> 0.99$ |
| $10^{-3}$ | 0.99 | $> 0.99$ |
| $10^{-2}$ | 0.89 | 0.98 |
| $10^{-1}$ | 0.21 | 0.19 |

As shown in Tab. 1, the actual updates are closely tracked by the first-order prediction up to $\eta = 10^{-2}$, with slopes near one and high $R^2$ values. The approximation breaks down at $\eta = 10^{-1}$, indicating that the update is leaving the local linear regime. These results support the practical relevance of analyzing logical propagation in the first-order geometry for local LLM updates.

# B. Primer on Finite Group Representations

This section collects basic definitions from group theory (Dummit & Foote, 2003) and group representation theory (Serre et al., 1977) used in our proofs. We work over real vector spaces unless otherwise stated, and all groups considered in this paper are finite.

## B.1. Groups

A *group* is a set $G$ equipped with a binary operation $(g, h) \mapsto gh$ that encodes the structure of symmetries. Formally, it must satisfy three axioms:

- **Associativity:** $(gh)k = g(hk)$ for all $g, h, k \in G$.

- **Identity:** There exists an element $e \in G$ such that $eg = ge = g$ for all $g$.

- **Inverses:** For every $g \in G$, there exists an element $g^{-1} \in G$ such that $gg^{-1} = g^{-1}g = e$.

We introduce two standard families of groups:

- **Symmetric group** $(S_n)$**:** The set of all permutations (bijections) of a set of $n$ elements forms a group under function composition.

- **Cyclic group** $(\mathbb{Z}_n)$**:** A group generated by a single element $g$ such that $g^n = e$ (and $g^k \neq e$ for $k < n$). We specifically use the cyclic group of order 2, $\mathbb{Z}_2 \cong (\{+1, -1\}, \times)$, to model logical negation, where the non-identity element corresponds to the negation operator $((-1)^2 = 1)$.

## B.2. Group Actions

A (left) *action* of a group $G$ on a set $X$ formalizes the idea of transforming elements of $X$ using the symmetries in $G$. It is a map $G \times X \to X$, denoted $(g, x) \mapsto g \cdot x$, that obeys two consistency rules:

- **Identity preservation:** $e \cdot x = x$ for all $x \in X$ (the identity transformation leaves everything unchanged).

- **Compatibility:** $g \cdot (h \cdot x) = (gh) \cdot x$ for all $g, h \in G$ and $x \in X$ (transforming by $h$ then $g$ is equivalent to transforming by the combined element $gh$).

Given an element $x \in X$, its *orbit* is the set of all points reachable from $x$ under the group action: $G \cdot x := \{g \cdot x : g \in G\}$.

## B.3. Product Groups

Let $G$ and $H$ be finite groups. The *direct product* $G \times H$ is the group of pairs $(g, h)$ defined by the component-wise operation:
$$(g, h)(g', h') = (gg', hh').$$

This structure allows us to compose symmetries acting on different domains. Specifically, if $G$ acts on a set $X$ and $H$ acts on a set $Y$, then the product group $G \times H$ naturally acts on the product set $X \times Y$ via

$$(g, h) \cdot (x, y) := (g \cdot x, h \cdot y).$$

## B.4. Representations and Equivariant Maps

Let $W$ be a real vector space. A (finite-dimensional) *representation* of $G$ on $W$ is a group homomorphism

$$\rho : G \to GL(W),$$

where $GL(W)$ denotes the group of invertible linear maps $W \to W$. This definition allows us to view $G$ as *acting linearly* on the vector space $W$. We denote this action by $g \cdot w := \rho(g)w$. Since each $\rho(g)$ lies in $GL(W)$, this action inherently preserves the linear structure: $g \cdot (aw + bv) = a(g \cdot w) + b(g \cdot v)$ for all scalars $a, b$ and vectors $w, v$.

A linear map $T : W \to W'$ between two $G$-representations $(W, \rho)$ and $(W', \rho')$ is called *G-equivariant* if it commutes with the group action:
$$T(g \cdot w) = g \cdot T(w) \qquad \text{for all } g \in G, \ w \in W.$$

In terms of the homomorphism $\rho$, this is equivalent to $T(\rho(g)w) = \rho'(g)T(w)$. The space of all such $G$-equivariant linear maps is denoted $\text{Hom}_G(W, W')$.

## B.5. Invariant Subspaces and Irreducibility

A subspace $U \subseteq W$ is called *G-invariant* if it is closed under the group action:

$$\rho(g)u \in U \quad \text{for all } g \in G, u \in U.$$

Intuitively, if a feature vector lies in a $G$-invariant subspace, applying any symmetry from $G$ will keep the vector within that same subspace. In this case, the restriction $\rho|_U : G \to GL(U)$ defines a valid representation on $U$, called a *subrepresentation*.

A representation is *irreducible* if it contains no proper nontrivial invariant subspaces. That is, the only $G$-invariant subspaces are $\{0\}$ and $W$ itself. A representation is *reducible* if it has a nontrivial invariant subspace.

A representation $W$ is *completely reducible* if it can be decomposed entirely into these atomic blocks. Formally, this means $W$ is isomorphic to a direct sum of irreducible subrepresentations:

$$W \cong \bigoplus_{i=1}^{\ell} W_i,$$

where each $W_i$ is irreducible.

### B.6. Maschke's Theorem

A key structural fact we use is that finite-dimensional representations of finite groups over $\mathbb{R}$ are completely reducible (Maschke's theorem). While the theorem holds for more general fields, we focus on the real field relevant to our work.

**Theorem 4** (Maschke). *Let $G$ be a finite group and let $W$ be a finite-dimensional representation of $G$ over $\mathbb{R}$. Then $W$ is completely reducible: every $G$-invariant subspace $U \subseteq W$ admits a $G$-invariant complement $U'$ such that $W = U \oplus U'$.*

We note that by recursively applying this property, i.e., finding an invariant subspace, splitting it off, and repeating the process on the complement, one can guarantee that any finite-dimensional representation $W$ can be fully decomposed into a direct sum of irreducible subrepresentations.

## C. Proof of Theorem 1

In this section, we provide the complete proof of Theorem 1. We proceed in four logical steps:

1. **Geometric Constraints:** First, we analyze the linear dependencies in the feature space, showing that the kernel of the feature map decomposes according to logical families under SLP (Lemma 3).

2. **Group Action Construction:** Building on this kernel structure, we translate the symbolic symmetries of entity renaming and negation to a well-defined linear representation of the group $H$ on the feature space (Lemmas 6 and 7).

3. **Representation Theoretic Decomposition:** We invoke standard results from representation theory to decompose the feature space into a direct sum of tensor products (Lemma 8).

4. **Derivation of Factorization:** Finally, we utilize the isomorphism of equivariant maps on tensor products (Lemma 9) to derive the explicit context-relation factorization of the feature map (Theorem 1).

### C.1. Logical Families and Controllability

Let $V$ be the free real vector space with basis $\{e_q : q \in Q\}$, where $e_q$ is a basis vector corresponding to the query $q$. Define a linear map

$$\Phi : V \longrightarrow W, \qquad \Phi(e_q) := \phi_q.$$

By construction, $\ker \Phi$ is the space of all linear dependencies among the feature vectors $\{\phi_q\}$. Moreover, recall from Sec. 2.4 that the two logical operations in the unary relation algebra generate a group $G := \{\text{id}, \neg, \text{rev}, \neg \circ \text{rev}\}$ acting on $Q$, and logical family is defined as:

$$G \cdot q := \{g(q) : g \in G\}.$$

Intuitively, the second condition of SLP (Def. 3) states that no unintended collapse happens *across* different logical families. In other words, linear dependencies arise if and only if they are semantically forced *within* a family. The following lemma states this formally.

**Lemma 3** (Family-wise kernel decomposition). *Let $F$ be a logical family and assume SLP (Def. 3). Let $V(F) := \text{span}\{e_q : q \in F\} \subseteq V$ and let $\Phi|_{V(F)}$ denote the restriction of $\Phi$ to $V(F)$. Then any linear relation among features decomposes family-wise, i.e.,*

$$\ker \Phi = \bigoplus_F \ker \Phi|_{V(F)}.$$

*Proof.* The logical families $\{F\}$ form a partition of $Q$, so the basis $\{e_q : q \in Q\}$ of $V$ splits accordingly into disjoint subsets $\{e_q : q \in F\}$. Hence

$$V = \bigoplus_F V(F),$$

and every $v \in V$ can be written uniquely as $v = \sum_F v_F$ with $v_F \in V(F)$.

By logical equivariance (Def. 2), for each family $F$, we can choose a representative $q_F^* \in F$ such that, for every $q \in F$, the feature vector $\phi_q$ is either $\phi_{q_F^*}$ or $-\phi_{q_F^*}$. Equivalently, for each $q \in F$ there exists a scalar $\lambda_{q,q^*} \in \{+1, -1\}$ with

$$\phi_q = \lambda_{q,q^*}\, \phi_{q_F^*}.$$

By the second condition of SLP (Def. 3), the chosen representatives can be taken so that the set $\{\phi_{q_F^*} : F$ is a logical family$\}$ is linearly independent in $W$.

Now take any $v \in \ker \Phi$ and decompose it as $v = \sum_F v_F$ with $v_F \in V(F)$. Write each component as

$$v_F = \sum_{q \in F} \alpha_q \, e_q.$$

Then

$$\Phi(v_F) = \sum_{q \in F} \alpha_q \, \phi_q = \sum_{q \in F} \alpha_q \, \lambda_{q,q^*} \, \phi_{q_F^*} = \beta_F \, \phi_{q_F^*},$$

where we define

$$\beta_F := \sum_{q \in F} \alpha_q \, \lambda_{q,q^*}.$$

Since $v \in \ker \Phi$, we have

$$0 = \Phi(v) = \sum_F \Phi(v_F) = \sum_F \beta_F \, \phi_{q_F^*}.$$

The vectors $\{\phi_{q_F^*}\}_F$ are linearly independent, so each coefficient must vanish:

$$\beta_F = 0 \quad \text{for all } F.$$

Hence $\Phi(v_F) = \beta_F \phi_{q_F^*} = 0$ for every family $F$, i.e. $v_F \in \ker \Phi|_{V(F)}$. Since $v = \sum_F v_F$ and the decomposition $V = \bigoplus_F V(F)$ is unique, this shows that every element of $\ker \Phi$ decomposes uniquely as a sum of elements from the subspaces $\ker \Phi|_{V(F)}$, and therefore

$$\ker \Phi = \bigoplus_F \ker \Phi|_{V(F)}.$$

$\square$

### C.2. Diagonal Renaming of Entities

We now formalize the idea that logical notions should be insensitive to the names of entities.

**Definition 5** (Entity renaming). *Let $G_E := \mathrm{Sym}(E)$ be the permutation group of E. We let $G_E$ act on Q by renaming entities uniformly in both argument slots:*

$$\sigma \cdot (h, r, t) := (\sigma(h), \, r, \, \sigma(t)) \quad (\sigma \in G_E, \, (h, r, t) \in Q).$$

An important observation is that, since entity renaming permutes only entities and logical operations act only on relations, **the two operations commute**.

**Lemma 4** (Symbolic renaming invariance). *For any $\sigma \in G_E$ and any $q \in Q$ we have*

$$\neg(\sigma \cdot q) = \sigma \cdot \neg(q), \qquad \mathrm{rev}(\sigma \cdot q) = \sigma \cdot \mathrm{rev}(q).$$

*Proof.* This is the direct consequence of the definition of logical operations in the tiny relation algebra and Def. 5. $\square$

In other words, if two queries are in the same logical family, then their renamings are in the same family.

Our goal is to show that this renaming symmetry *translates* to a linear symmetry of $W$. To this end, we first show that the kernel structure is preserved upon entity renaming. For each $\sigma \in G_E$ define a linear map $P_\sigma : V \to V$ on basis vectors by

$$P_\sigma e_q := e_{\sigma \cdot q}.$$

**Lemma 5** (Kernel invariance under renaming). *For every $\sigma \in G_E$, if $v \in \ker \Phi$, then $P_\sigma v \in \ker \Phi$.*

*Proof.* Let $v \in \ker \Phi$. By Lemma 3, it suffices to show the invariance for a component $v_F \in \ker \Phi|_{V(F)}$ entirely contained in a single logical family $F$. By the logical equivariance assumption, the feature vectors within a family $F$ are all collinear. Specifically, fix a representative $q^* \in F$. Then for any $q \in F$, there exists $\lambda_{q,q^*} \in \{+1, -1\}$ such that

$$\phi_q = \lambda_{q,q^*} \phi_{q^*}.$$

Consequently, a vector $v_F = \sum_{q \in F} \alpha_q e_q$ lies in $\ker \Phi$ if and only if

$$\Phi(v_F) = \sum_{q \in F} \alpha_q \phi_q = \left( \sum_{q \in F} \alpha_q \lambda_{q,q^*} \right) \phi_{q^*} = 0.$$

Since $\phi_{q^*} \neq 0$ (by the second condition of SLP Def. 3), the condition for membership in the kernel is purely scalar:

$$\sum_{q \in F} \alpha_q \lambda_{q,q^*} = 0. \tag{8}$$

Now consider the transformed vector $P_\sigma v_F = \sum_{q \in F} \alpha_q e_{\sigma \cdot q}$. This vector is supported on the permuted family $\sigma(F)$. Let $p^* := \sigma \cdot q^*$ be the representative for $\sigma(F)$. For any $q \in F$, let $p := \sigma \cdot q$. By definition of a logical family (orbit under the logical-operation group $G$ generated by $\neg$ and rev), for any $q \in F$ there exists $g \in G$ such that

$$q = g \cdot q^*.$$

We claim that entity renaming commutes with every $g \in G$:

$$\sigma \cdot (g \cdot x) = g \cdot (\sigma \cdot x) \qquad \forall \sigma \in G_E, \ \forall g \in G, \ \forall x \in Q. \tag{9}$$

Indeed, Lemma 4 gives this commutation for the generators $\neg$ and rev, and Eq. (9) follows for arbitrary $g$ by closure under composition in $G$.

Applying Eq. (9) to $x = q^*$ yields

$$p := \sigma \cdot q = \sigma \cdot (g \cdot q^*) = g \cdot (\sigma \cdot q^*) =: g \cdot p^*,$$

so the same $g$ relates $p$ to $p^*$.

Next, by the logical equivariance condition, there is a sign homomorphism $\chi : G \to \{+1, -1\}$ such that for all $x \in Q$,

$$\phi_{g \cdot x} = \chi(g) \phi_x. \tag{10}$$

Using Eq. (10) with $x = q^*$ and $x = p^*$, we relate the scalar coefficients to $\chi$:

$$\phi_q = \phi_{g \cdot q^*} = \chi(g)\phi_{q^*} \quad \Longrightarrow \quad \lambda_{q,q^*} = \chi(g),$$

$$\phi_p = \phi_{g \cdot p^*} = \chi(g)\phi_{p^*} \quad \Longrightarrow \quad \lambda_{p,p^*} = \chi(g).$$

Therefore, the relative signs are preserved:

$$\lambda_{p,p^*} = \chi(g) = \lambda_{q,q^*}.$$

Using the equality above, we finally show that $P_\sigma v_F \in \ker \Phi$:

$$\begin{aligned}
\Phi(P_\sigma v_F) &= \sum_{q \in F} \alpha_q \phi_{\sigma \cdot q} \\
&= \sum_{q \in F} \alpha_q \lambda_{\sigma \cdot q, \sigma \cdot q^*} \phi_{p^*} \\
&= \left( \sum_{q \in F} \alpha_q \lambda_{q,q^*} \right) \phi_{p^*}.
\end{aligned}$$

By equation Eq. (8), the coefficient sum is zero. Thus $\Phi(P_\sigma v_F) = 0$, proving that $P_\sigma v_F \in \ker \Phi$. $\qquad \square$

Using Lemma 5, we can translate the entity renaming to a linear symmetry on the feature space.

**Lemma 6.** *Suppose that for each $\sigma \in G_E$ we have $P_\sigma(\ker \Phi) \subseteq \ker \Phi$. Then there exists a unique linear map $\rho_E(\sigma) : W \to W$ such that*

$$\phi_{\sigma \cdot q} = \rho_E(\sigma) \phi_q \quad \forall q \in Q.$$

*Moreover, the assignment $\sigma \mapsto \rho_E(\sigma)$ defines a group representation $\rho_E : G_E \to GL(W)$.*

*Proof.* Recall that $\Phi : V \to W$ is surjective by definition of $W = \text{span}\{\phi_q : q \in Q\}$. For a fixed $\sigma \in G_E$, we define $\rho_E(\sigma)$ as follows: given any $w \in W$, choose $v \in V$ with $\Phi(v) = w$ and set

$$\rho_E(\sigma) w := \Phi(P_\sigma v).$$

We first check that this does not depend on the choice of $v$. Suppose $v, v' \in V$ satisfy $\Phi(v) = \Phi(v')$, i.e. $\Phi(v - v') = 0$, so $v - v' \in \ker \Phi$. By assumption, $P_\sigma(\ker \Phi) \subseteq \ker \Phi$, hence

$$\Phi(P_\sigma(v - v')) = 0,$$

which implies

$$\Phi(P_\sigma v) = \Phi(P_\sigma v').$$

Thus, $\rho_E(\sigma) w$ is well-defined. Linearity of $\rho_E(\sigma)$ follows immediately from the linearity of $\Phi$ and $P_\sigma$.

Now, we apply this definition to the basis elements. For $q \in Q$, we have $e_q \in V$ and $\Phi(e_q) = \phi_q$, so

$$\rho_E(\sigma) \phi_q = \rho_E(\sigma) \Phi(e_q) = \Phi(P_\sigma e_q) = \Phi(e_{\sigma \cdot q}) = \phi_{\sigma \cdot q}.$$

To check its uniqueness, observe that the vectors $\{\phi_q : q \in Q\}$ span $W$, so any linear map $T : W \to W$ satisfying $T \phi_q = \phi_{\sigma \cdot q}$ for all $q$ must agree with $\rho_E(\sigma)$ on a spanning set, hence must be equal to $\rho_E(\sigma)$.

Finally, we verify the group property. For any $\sigma_1, \sigma_2 \in G_E$ and any, $w = \Phi(v)$ we have

$$\rho_E(\sigma_1) \rho_E(\sigma_2) w = \rho_E(\sigma_1) \Phi(P_{\sigma_2} v) = \Phi(P_{\sigma_1} P_{\sigma_2} v) = \Phi(P_{\sigma_1 \sigma_2} v) = \rho_E(\sigma_1 \sigma_2) w.$$

Thus, $\rho_E(\sigma_1)\rho_E(\sigma_2) = \rho_E(\sigma_1 \sigma_2)$ on all of $W$, and $\rho_E : G_E \to GL(W)$ is a representation. $\square$

Hence, SLP implies that the linearized features carry a compatible entity-renaming symmetry.

### C.3. Combining Renaming and Negation

We now combine entity renaming with relation negation into a single symmetry structure. Recall from Def. 2 that logical equivariance of negation says

$$\phi_{\neg q} = - \phi_q \quad \forall q \in Q,$$

i.e., relation-level negation always flips the feature vector by a global sign $-1$, regardless of which entities appear.

Entity renaming acts only on the head and tail slots, while negation acts only on the relation slot. We want to treat these two operations together as a single family of joint transformations.

**Definition 6** (Combined symmetry group). *Let $\mathbb{Z}_2 := \{+1, -1\}$ be the two-element group with multiplication. We define*

$$H := G_E \times \mathbb{Z}_2.$$

*An element $(\sigma, \epsilon) \in H$ acts on a query by*

$$(\sigma, \epsilon) \cdot (h, r, t) := (\sigma(h), \epsilon \cdot r, \sigma(t)),$$

*where $\epsilon = +1$ means "keep the relation as it is" and $\epsilon = -1$ means "replace $r$ by its negation $\neg r$".*

*On the feature space $W$ we let $(\sigma, \epsilon)$ act linearly by*

$$\rho(\sigma, +1) := \rho_E(\sigma), \qquad \rho(\sigma, -1) := - \rho_E(\sigma),$$

*so that $\rho(\sigma, \epsilon)$ first applies the renaming operator $\rho_E(\sigma)$ and then, if $\epsilon = -1$, flips the sign of the feature vector. In other words, for any $w \in W$,*

$$\rho(\sigma, \epsilon)\, w := \begin{cases} \rho_E(\sigma)\, w & \text{if } \epsilon = +1, \\ -\rho_E(\sigma)\, w & \text{if } \epsilon = -1. \end{cases}$$

Because $\rho_E$ respects composition of renamings and $(-1)^2 = +1$, the maps $\rho(\sigma, \epsilon)$ also respect composition: applying $(\sigma_1, \epsilon_1)$ and then $(\sigma_2, \epsilon_2)$ has the same effect on features as applying $(\sigma_1 \sigma_2, \epsilon_1 \epsilon_2)$ once. This means $H$ acts consistently and linearly on $W$. Hence, logical equivariance and renaming equivariance can now be combined into a single statement as below.

**Lemma 7** (Equivariant feature map). *The map $\phi : Q \to W$ is compatible with the combined symmetry action of $H$ in the sense that, for all $(\sigma, \epsilon) \in H$ and all $q \in Q$,*

$$\phi\big((\sigma, \epsilon) \cdot q\big) = \rho(\sigma, \epsilon)\, \phi_q.$$

*Proof.* When $\epsilon = +1$, we have

$$(\sigma, +1) \cdot (h, r, t) = (\sigma(h), r, \sigma(t)),$$

and this is represented on features by $\rho_E(\sigma)$, i.e. $\phi(\sigma \cdot q) = \rho_E(\sigma)\, \phi_q = \rho(\sigma, +1)\, \phi_q$.

When $\epsilon = -1$, we have

$$(\sigma, -1) \cdot (h, r, t) = (\sigma(h), \neg r, \sigma(t)).$$

By logical equivariance of negation, $\phi_{\neg q} = -\phi_q$ for every $q$, and by Lemma 4, negation commutes with renaming on the symbolic side. Combining these facts, we obtain

$$\phi\big((\sigma, -1) \cdot q\big) = \phi\big(\sigma \cdot (\neg q)\big) = \rho_E(\sigma)\, \phi_{\neg q} = \rho_E(\sigma)\, (-\phi_q) = -\rho_E(\sigma)\, \phi_q = \rho(\sigma, -1)\, \phi_q,$$

as claimed. $\qquad\square$

In words, performing a renaming and optional negation on the symbolic query side has the same effect as applying the corresponding linear operator $\rho(\sigma, \epsilon)$ on the feature side.

### C.4. Decomposition of $H$-representations

Using the Lemmas above, we prove Lemma 8, which will play a crucial role in proving Theorem 1.

**Lemma 8** (Decomposition of $H$-representations). *Let $W$ be a finite-dimensional real representation of $H = G_E \times \mathbb{Z}_2$. Then $W$ decomposes into a direct sum of tensor-product blocks:*

$$W \cong \bigoplus_{i=1}^{m} (C_i \otimes R_i),$$

*where each $C_i$ is an irreducible representation of the entity group $G_E$, and each $R_i$ is an irreducible representation of the logical group $\mathbb{Z}_2$.*

*Proof.* We explicitly construct the decomposition using the structure of $\mathbb{Z}_2 = \{1, -1\}$. Let $z := (\text{id}, -1) \in H$. Since $z$ commutes with every element in $H$ (i.e., $zh = hz$ for all $h$), the linear map $\rho(z)$ must commute with the group action:

$$\rho(z)\rho(h) = \rho(zh) = \rho(hz) = \rho(h)\rho(z) \quad \text{for all } h \in H.$$

Also, since $z^2 = (\text{id}, 1) = e$, we have $\rho(z)^2 = I$. Thus, $\rho(z)$ is an involution and $W$ can be decomposed into eigenspaces corresponding to eigenvalues $+1$ and $-1$:

$$W = W^+ \oplus W^-, \quad \text{where } W^\pm := \{w \in W : \rho(z)w = \pm w\}.$$

Crucially, these eigenspaces are $H$-invariant. To see this, let $w \in W^\pm$ and $h \in H$. Using the commutativity shown above:

$$\rho(z)\big(\rho(h)w\big) = \rho(h)\big(\rho(z)w\big) = \rho(h)(\pm w) = \pm\big(\rho(h)w\big).$$

This shows that if $w \in W^{\pm}$, then the transformed vector $\rho(h)w$ also satisfies the condition to be in $W^{\pm}$. Thus, $W^+$ and $W^-$ are subrepresentations of $H$.

We now determine the action of a general element $(\sigma, \epsilon) \in H$ on these subspaces. Note that, any element factors as $(\sigma, \epsilon) = (\sigma, 1)(\text{id}, \epsilon)$. Then, the logical factor $(\text{id}, \epsilon)$ acts on the subspaces as:

$$\rho(\text{id}, \epsilon)\big|_{W^+} = I, \qquad \rho(\text{id}, \epsilon)\big|_{W^-} = \epsilon I.$$

Consequently, the full action is:

$$\rho(\sigma, \epsilon)w \;=\; \rho(\sigma, 1)\rho(\text{id}, \epsilon)w \;=\; \begin{cases} \rho(\sigma, 1)w & \text{if } w \in W^+, \\ \epsilon \cdot \rho(\sigma, 1)w & \text{if } w \in W^-. \end{cases}$$

Since $W$ is finite-dimensional, we can apply Maschke's Theorem (Theorem 4) to decompose $W^+$ and $W^-$ into irreducible representations of $G_E$ (via the map $\sigma \mapsto \rho(\sigma, 1)$):

$$W^+ \cong \bigoplus_{j \in J_+} C_j, \qquad W^- \cong \bigoplus_{j \in J_-} C_j.$$

Finally, we map this structure to the tensor product form claimed in the theorem. Let $\mathbf{1}$ and $\text{sgn}$ denote the two irreducible real representations of $\mathbb{Z}_2$, respectively. Each irreducible $G_E$-summand $C \subseteq W^+$ is an $H$-subrepresentation on which $(\sigma, \epsilon)$ acts by $\rho(\sigma, 1)$ alone, hence it is $H$-isomorphic to $C \otimes \mathbf{1}$ (where $(\sigma, \epsilon) \cdot (c \otimes 1) = (\sigma c) \otimes 1$). Likewise, each irreducible $G_E$-summand $C \subseteq W^-$ is an $H$-subrepresentation on which $(\sigma, \epsilon)$ acts by $\epsilon \, \rho(\sigma, 1)$, hence it is $H$-isomorphic to $C \otimes \text{sgn}$ (where $(\sigma, \epsilon) \cdot (c \otimes 1) = (\sigma c) \otimes \epsilon$). Therefore,

$$W \;\cong\; \left( \bigoplus_{j \in J_+} C_j \otimes \mathbf{1} \right) \;\oplus\; \left( \bigoplus_{j \in J_-} C_j \otimes \text{sgn} \right).$$

Letting $R_i$ represent either $\mathbf{1}$ or $\text{sgn}$ as appropriate for each block, we obtain the claimed form $W \cong \bigoplus_i (C_i \otimes R_i)$. $\qquad \square$

### C.5. Equivariant Maps on Tensor Products

To prove the main theorem, we need to characterize equivariant maps between tensor product representations.

**Lemma 9** (Equivariant maps on tensor products). *Let $G$ and $K$ be groups. Let $U, A$ be finite-dimensional real representations of $G$, and let $V, B$ be finite-dimensional real representations of $K$. We regard the tensor products $U \otimes V$ and $A \otimes B$ as representations of the product group $G \times K$ via the component-wise action:*

$$(g, k) \cdot (u \otimes v) = (g \cdot u) \otimes (k \cdot v), \quad (g, k) \cdot (a \otimes b) = (g \cdot a) \otimes (k \cdot b).$$

*Then there is a natural linear isomorphism*

$$\text{Hom}_G(U, A) \otimes \text{Hom}_K(V, B) \;\cong\; \text{Hom}_{G \times K}(U \otimes V, \, A \otimes B).$$

*Proof.* Consider first the full vector spaces of linear maps without equivariance constraints. For any linear maps $f \in \text{Hom}(U, A)$ and $h \in \text{Hom}(V, B)$, we define a map $f \boxtimes h : U \otimes V \to A \otimes B$ by its action on simple tensors:

$$(f \boxtimes h)(u \otimes v) := f(u) \otimes h(v) \quad \text{for all } u \in U, v \in V.$$

This construction induces a natural linear map

$$\Psi : \text{Hom}(U, A) \otimes \text{Hom}(V, B) \to \text{Hom}(U \otimes V, A \otimes B), \qquad \Psi(f \otimes h) = f \boxtimes h.$$

We first claim that $\Psi$ is an isomorphism. To see this, choose bases for the source and target vector spaces:

$$(u_i)_{i=1}^m \text{ of } U, \qquad\qquad (a_j)_{j=1}^p \text{ of } A,$$
$$(v_r)_{r=1}^n \text{ of } V, \qquad\qquad (b_s)_{s=1}^q \text{ of } B.$$

We define the elementary linear maps that form the bases for the hom-spaces. For each pair $(j, i)$, let $E_{j,i} \in \mathrm{Hom}(U, A)$ be the unique linear map defined by:

$$E_{j,i}(u_k) = \begin{cases} a_j & \text{if } k = i, \\ 0 & \text{if } k \neq i. \end{cases}$$

The set $\{E_{j,i}\}_{j,i}$ forms a basis of $\mathrm{Hom}(U, A)$.

Similarly, for each $(s, r)$, let $F_{s,r} \in \mathrm{Hom}(V, B)$ be defined by:

$$F_{s,r}(v_k) = \begin{cases} b_s & \text{if } k = r, \\ 0 & \text{if } k \neq r. \end{cases}$$

The set $\{F_{s,r}\}_{s,r}$ forms a basis of $\mathrm{Hom}(V, B)$. Consequently, the set of tensor products forms a basis for the domain of $\Psi$:

$$\{E_{j,i} \otimes F_{s,r}\}_{j,i,s,r} \quad \subset \quad \mathrm{Hom}(U, A) \otimes \mathrm{Hom}(V, B).$$

Now, consider the image of these basis vectors under $\Psi$. Let $M_{j,i,s,r} := \Psi(E_{j,i} \otimes F_{s,r})$. By the definition of $\Psi$ (action on simple tensors), $M_{j,i,s,r}$ is the unique linear map $U \otimes V \to A \otimes B$ satisfying:

$$M_{j,i,s,r}(u_{i'} \otimes v_{r'}) = E_{j,i}(u_{i'}) \otimes F_{s,r}(v_{r'}) = \begin{cases} a_j \otimes b_s & \text{if } (i', r') = (i, r), \\ 0 & \text{otherwise.} \end{cases}$$

The collection of maps $\{M_{j,i,s,r}\}_{j,i,s,r}$ is precisely the standard basis of the codomain space $\mathrm{Hom}(U \otimes V, A \otimes B)$. Since $\Psi$ maps a basis of the domain bijectively onto a basis of the codomain, it is a linear isomorphism.

Our strategy is to identify equivariant maps as the fixed points of a group action on the function space. Specifically, we equip $\mathrm{Hom}(U, A)$ with a natural $G$-action via conjugation: for $g \in G$ and $f \in \mathrm{Hom}(U, A)$, the transformed map $g \cdot f$ is defined by

$$(g \cdot f)(u) := g \cdot f(g^{-1} \cdot u).$$

A map $f$ is $G$-equivariant if and only if it is invariant under this action (i.e., $g \cdot f = f$ for all $g$), hence

$$\mathrm{Hom}_G(U, A) = \mathrm{Hom}(U, A)^G,$$

where the superscript $G$ denotes the subspace of fixed points. Analogously, we define the $K$-action on $\mathrm{Hom}(V, B)$, and the $(G \times K)$-action on $\mathrm{Hom}(U \otimes V, A \otimes B)$: for any $T \in \mathrm{Hom}(U \otimes V, A \otimes B)$, the action is defined by

$$\big((g, k) \cdot T\big)(x) := (g, k) \cdot T\big((g, k)^{-1} \cdot x\big).$$

We now verify that the isomorphism $\Psi$ respects these group structures. We equip the domain, $\mathrm{Hom}(U, A) \otimes \mathrm{Hom}(V, B)$, with the component-wise action of $G \times K$:

$$(g, k) \cdot (f \otimes h) := (g \cdot f) \otimes (k \cdot h).$$

For any $(g, k) \in G \times K$, a direct computation on simple tensors $u \otimes v$ shows:

$$\begin{aligned}
\big((g, k) \cdot (f \boxtimes h)\big)(u \otimes v) &= (g, k) \cdot (f \boxtimes h)\big((g, k)^{-1} \cdot (u \otimes v)\big) \\
&= (g, k) \cdot (f \boxtimes h)(g^{-1}u \otimes k^{-1}v) \\
&= (g, k) \cdot \big(f(g^{-1}u) \otimes h(k^{-1}v)\big) \\
&= (g \cdot f(g^{-1}u)) \otimes (k \cdot h(k^{-1}v)) \\
&= \big((g \cdot f) \boxtimes (k \cdot h)\big)(u \otimes v).
\end{aligned}$$

Since this equality holds for all simple tensors $u \otimes v$, which span $U \otimes V$, the linear maps are identical. Rewriting this using $\Psi$, we see that:

$$(g, k) \cdot \Psi(f \otimes h) = \Psi\big((g, k) \cdot (f \otimes h)\big).$$

Therefore, $\Psi$ restricts to an isomorphism between the respective fixed-point subspaces:

$$\Psi : (\mathrm{Hom}(U, A) \otimes \mathrm{Hom}(V, B))^{G \times K} \xrightarrow{\sim} \mathrm{Hom}(U \otimes V, A \otimes B)^{G \times K}.$$

Finally, we identify the invariants of the tensor product. Let $X$ be any $G$-representation and $Y$ any $K$-representation, and consider $X \otimes Y$ as a $(G \times K)$-representation via $(g, k) \cdot (x \otimes y) = (g \cdot x) \otimes (k \cdot y)$. Then $(X \otimes Y)^{G \times K} = X^G \otimes Y^K$. Indeed, if $w \in (X \otimes Y)^{G \times K}$, then in particular $w \in (X \otimes Y)^{\{e\} \times K}$, so $w \in X \otimes Y^K$. Write $w = \sum_i x_i \otimes y_i$ with $y_i \in Y^K$. Now, invariance under $G \times \{e\}$ implies

$$\sum_i (g \cdot x_i) \otimes y_i = \sum_i x_i \otimes y_i \quad \text{for all } g \in G,$$

and since the $y_i$ lie in $Y^K$, we may choose them to be linearly independent by regrouping terms. It follows that $g \cdot x_i = x_i$ for each $i$, hence $x_i \in X^G$ and therefore $w \in X^G \otimes Y^K$. The reverse inclusion $X^G \otimes Y^K \subseteq (X \otimes Y)^{G \times K}$ is immediate, proving the claim.

Applying this with $X = \mathrm{Hom}(U, A)$ and $Y = \mathrm{Hom}(V, B)$ gives

$$(\mathrm{Hom}(U, A) \otimes \mathrm{Hom}(V, B))^{G \times K} \;=\; \mathrm{Hom}(U, A)^G \otimes \mathrm{Hom}(V, B)^K \;=\; \mathrm{Hom}_G(U, A) \otimes \mathrm{Hom}_K(V, B).$$

Combining this with the fact that $\Psi$ restricts to an isomorphism on the invariant subspaces, we conclude:

$$\mathrm{Hom}_G(U, A) \otimes \mathrm{Hom}_K(V, B) \;\cong\; \mathrm{Hom}_{G \times K}(U \otimes V, A \otimes B).$$

$\square$

### C.6. Proof of Theorem 1

Lemma 8 implies that, with a proper choice of basis, any feature vector $\phi_q$ in a model satisfying SLP can be expressed as a direct sum of components, where each component is a tensor product of an entity-dependent factor (from $C_i$) and a relation-dependent factor (from $R_i$). We now translate this general decomposition into specific constraints on the query feature map $\phi(h, r, t)$, to prove Theorem 1.

**Theorem 1** (Context-Relation Factorization). *Let $\phi : Q \to W$ be the feature map defined by $q \mapsto \phi_q$. If $\phi$ satisfies SLP, then there exist real vector spaces $\{C_i\}_i$, $\{R_i\}_i$ and an isomorphism $W \cong \bigoplus_i (C_i \otimes R_i)$ such that*

$$\phi(h, r, t) \;=\; \bigoplus_i \left( \sum_{k=1}^{m_i} u_{i,k}(h, t) \otimes v_{i,k}(r) \right),$$

*where $u_{i,k} : E \times E \to C_i$ and $v_{i,k} : \mathcal{R} \to R_i$. Moreover, negation acts locally as a sign flip on each relation component, i.e., $v_{i,k}(\neg r) = -v_{i,k}(r)$.*[6]

*Proof.* Let $V$ with the free vector space with basis $\{e_q : q \in Q\}$. The group action of $H$ on the set $Q$ naturally induces a linear representation on $V$, defined by permuting the basis vectors:

$$(\sigma, \epsilon) \cdot e_q \;:=\; e_{(\sigma, \epsilon) \cdot q}.$$

Let $\Phi : V \to W$ be the unique linear extension of $\phi$, defined by $\Phi(e_q) := \phi_q$. We identify $V$ with the tensor product

$$V \;\cong\; V_{\mathrm{ctx}} \otimes V_{\mathrm{rel}}$$

via the basis mapping $e_{(h,r,t)} \mapsto e_{(h,t)} \otimes e_r$. Under this identification, the $H$-action on $V$ acts component-wise:

$$(\sigma, \epsilon) \cdot (e_{(h,t)} \otimes e_r) \;:=\; e_{(\sigma(h), \sigma(t))} \otimes e_{\epsilon \cdot r}.$$

---

[6]We use $\bigoplus_i W_i$ for a direct-sum decomposition, meaning that a feature vector decomposes into separate components across the blocks $W_i$. We use $C \otimes R$ for a tensor product, which formalizes binding a context (entity-pair) factor in $C$ with a relation factor in $R$.

By Lemma 7, $\phi : Q \to W$ is $H$-equivariant with respect to the $H$-action on queries. Since $\Phi$ is defined linearly on the basis $Q$, this equivariance extends to the entire space $V$, making $\Phi$ an $H$-equivariant map. Moreover, by Lemma 8, one can fix an $H$-equivariant isomorphism $W \cong \bigoplus_i (C_i \otimes R_i)$, and let $\pi_i : W \to C_i \otimes R_i$ be the corresponding projection. Define the component map $\Phi_i := \pi_i \circ \Phi$. Since both $\Phi$ and $\pi_i$ are equivariant, $\Phi_i$ belongs to the space

$$\mathrm{Hom}_H(V_{\mathrm{ctx}} \otimes V_{\mathrm{rel}}, C_i \otimes R_i) \;\cong\; \mathrm{Hom}_{G_E}(V_{\mathrm{ctx}}, C_i) \otimes \mathrm{Hom}_{\mathbb{Z}_2}(V_{\mathrm{rel}}, R_i),$$

where the isomorphism is established by Lemma 9. Thus, $\Phi_i$ is an element of a tensor product space, implying it can be written as a finite sum of pure tensors:

$$\Phi_i \;=\; \sum_{k=1}^{m_i} U_{i,k} \otimes S_{i,k},$$

where $U_{i,k} : V_{\mathrm{ctx}} \to C_i$ is $G_E$-equivariant and $S_{i,k} : V_{\mathrm{rel}} \to R_i$ is $\mathbb{Z}_2$-equivariant. Defining the embeddings $u_{i,k}(h, t) := U_{i,k}(e_{(h,t)})$ and $v_{i,k}(r) := S_{i,k}(e_r)$, the overall feature map decomposes as:

$$\phi(h, r, t) \;=\; \bigoplus_i \left( \sum_{k=1}^{m_i} u_{i,k}(h, t) \otimes v_{i,k}(r) \right).$$

It remains to show that negation acts by sign on the relation embeddings. Let $\epsilon \in \mathbb{Z}_2$ be the negation element. The $H$-action on the feature space decomposes block-wise. On the $i$-th block $C_i \otimes R_i$, the action is component-wise:

$$(\mathrm{id}, \epsilon) \cdot (u \otimes v) \;=\; (\mathrm{id} \cdot u) \otimes (\epsilon \cdot v).$$

Since $R_i$ is a real irreducible representation of $\mathbb{Z}_2$, it is one-dimensional, so $\epsilon$ acts on $R_i$ as a scalar $\eta_i \in \{+1, -1\}$. Crucially, because the first component of the group element is the identity, the embedding $u_{i,k}$ remains unchanged. Thus, the action on the sum is:

$$\begin{aligned}
\phi_i(h, \neg r, t) &= (\mathrm{id}, \epsilon) \cdot \sum_{k=1}^{m_i} \big( u_{i,k}(h, t) \otimes v_{i,k}(r) \big) \\
&= \eta_i \sum_{k=1}^{m_i} \big( u_{i,k}(h, t) \otimes v_{i,k}(r) \big).
\end{aligned}$$

The SLP condition requires $\phi(h, \neg r, t) = -\phi(h, r, t)$ for all queries. Comparing this with the equation above, we see that for any block $i$ where the feature map is not identically zero, we must have $\eta_i = -1$. (If $\phi_i \equiv 0$, the SLP constraint is satisfied on this block for any $\eta_i$, hence the sign choice is immaterial.) Consequently, in all cases, the relation embeddings must satisfy the sign-flip property:

$$v_{i,k}(\neg r) \;=\; -v_{i,k}(r).$$

Thus, every contributing relation component transforms under negation by the sign representation. $\square$

## D. Proof of Theorem 2

In this section, we provide the proof of Theorem 2. We first prove the lemma below, which will play a crucial role in proving Theorem 2.

**Lemma 10** (Converse alignment in Hom-spaces). *Fix $i$ and an $H$-equivariant projection $\pi_i : W \to W_i \cong C_i \otimes R_i$. Let $\Phi_i := \pi_i \circ \Phi \in \mathrm{Hom}_H(V_{\mathrm{ctx}} \otimes V_{\mathrm{rel}}, C_i \otimes R_i)$, and identify this space with $\mathcal{A}_i \otimes \mathcal{B}_i$ via Lemma 9, where $\mathcal{A}_i = \mathrm{Hom}_{G_E}(V_{\mathrm{ctx}}, C_i)$ and $\mathcal{B}_i = \mathrm{Hom}_{\mathbb{Z}_2}(V_{\mathrm{rel}}, R_i)$. Let $P_{\mathrm{pair}} : V_{\mathrm{ctx}} \to V_{\mathrm{ctx}}$ and $P_{\mathrm{rel}} : V_{\mathrm{rel}} \to V_{\mathrm{rel}}$ be the linear maps defined by their action on the basis vectors:*

$$P_{\mathrm{pair}}(e_{(h,t)}) := e_{(t,h)} \qquad and \qquad P_{\mathrm{rel}}(e_r) := e_{r^\smile}.$$

*Note that the converse operation commutes with negation (i.e., $(\neg r)^\smile = \neg(r^\smile)$), so $P_{\mathrm{rel}}$ is $\mathbb{Z}_2$-equivariant. Define involutions $\mathcal{P}_{\mathrm{pair}}(U) = U \circ P_{\mathrm{pair}}$ on $\mathcal{A}_i$ and $\mathcal{P}_{\mathrm{rel}}(S) = S \circ P_{\mathrm{rel}}$ on $\mathcal{B}_i$. If $\phi(t, r^\smile, h) = \phi(h, r, t)$ for all $(h, r, t) \in Q$, then*

$$(\mathcal{P}_{\mathrm{pair}} \otimes \mathcal{P}_{\mathrm{rel}})(\Phi_i) = \Phi_i.$$

*Proof.* First, we verify that $\mathcal{P}_{\text{pair}}$ and $\mathcal{P}_{\text{rel}}$ are well-defined. For any $U \in \mathcal{A}_i$ and $g \in G_E$, since $P_{\text{pair}}$ commutes with the $G_E$-action,

$$(U \circ P_{\text{pair}})(g \cdot x) = U(P_{\text{pair}}(g \cdot x)) = U(g \cdot P_{\text{pair}}(x)) = g \cdot (U \circ P_{\text{pair}})(x),$$

so $\mathcal{P}_{\text{pair}}(U) \in \mathcal{A}_i$. Similarly, $\mathbb{Z}_2$-equivariance of $P_{\text{rel}}$ ensures $\mathcal{P}_{\text{rel}}$ is well-defined on $\mathcal{B}_i$.

Next, since $\phi(t, r^{\smile}, h) = \phi(h, r, t)$ for all $(h, r, t) \in Q$, $\Phi \circ (P_{\text{pair}} \otimes P_{\text{rel}}) = \Phi$. Applying $\pi_i$ gives

$$\Phi_i \circ (P_{\text{pair}} \otimes P_{\text{rel}}) = (\pi_i \circ \Phi) \circ (P_{\text{pair}} \otimes P_{\text{rel}}) = \pi_i \circ \Phi = \Phi_i.$$

Under the identification $\mathcal{A}_i \otimes \mathcal{B}_i \cong \text{Hom}_H(V_{\text{ctx}} \otimes V_{\text{rel}}, C_i \otimes R_i)$, a pure tensor $U \otimes S$ corresponds to the map $x \otimes y \mapsto U(x) \otimes S(y)$. Precomposing with $P_{\text{pair}} \otimes P_{\text{rel}}$ yields

$$(x \otimes y) \mapsto U(P_{\text{pair}}(x)) \otimes S(P_{\text{rel}}(y)) = (U \circ P_{\text{pair}})(x) \otimes (S \circ P_{\text{rel}})(y),$$

which is exactly the map corresponding to $(\mathcal{P}_{\text{pair}}U) \otimes (\mathcal{P}_{\text{rel}}S)$. Extending linearly shows that precomposition by $P_{\text{pair}} \otimes P_{\text{rel}}$ corresponds to $\mathcal{P}_{\text{pair}} \otimes \mathcal{P}_{\text{rel}}$, hence $(\mathcal{P}_{\text{pair}} \otimes \mathcal{P}_{\text{rel}})(\Phi_i) = \Phi_i$. $\qquad\square$

To prove Theorem 2, we restate its general form.

**Theorem 2** (Symmetric-Antisymmetric Alignment (Formal statement))**.** *Assume the context-relation factorization from Theorem 1 and $\phi(t, r^{\smile}, h) = \phi(h, r, t)$ for all $(h, r, t) \in Q$. Then, for each block $i$ (as in Theorem 1), the projected feature component $\phi_i := \pi_i \circ \phi : Q \to W_i \cong C_i \otimes R_i$ admits a decomposition*

$$\phi_i(h, r, t) = \phi_i^+(h, r, t) + \phi_i^-(h, r, t),$$

*where each part can be written as a finite sum of context–relation pure tensors*

$$\phi_i^{\pm}(h, r, t) = \sum_{k \in I_i^{\pm}} u_{i,k}^{\pm}(h, t) \otimes v_{i,k}^{\pm}(r),$$

*such that every summand satisfies the aligned symmetry relations*

$$u_{i,k}^{\pm}(t, h) = \pm u_{i,k}^{\pm}(h, t), \qquad v_{i,k}^{\pm}(r^{\smile}) = \pm v_{i,k}^{\pm}(r).$$

*Consequently,*

$$\phi(h, r, t) = \bigoplus_i \left( \phi_i^+(h, r, t) + \phi_i^-(h, r, t) \right)$$

*with the same componentwise symmetry properties on each block.*

*Proof.* Fix a block index $i$ and an $H$-equivariant projection $\pi_i : W \to W_i \cong C_i \otimes R_i$, and write $\Phi_i := \pi_i \circ \Phi \in \text{Hom}_H(V_{\text{ctx}} \otimes V_{\text{rel}}, C_i \otimes R_i)$. By Lemma 9, we identify

$$\text{Hom}_H(V_{\text{ctx}} \otimes V_{\text{rel}}, C_i \otimes R_i) \cong \mathcal{A}_i \otimes \mathcal{B}_i,$$

where $\mathcal{A}_i := \text{Hom}_{G_E}(V_{\text{ctx}}, C_i)$ and $\mathcal{B}_i := \text{Hom}_{\mathbb{Z}_2}(V_{\text{rel}}, R_i)$.

Let $\mathcal{P}_{\text{pair}}$ and $\mathcal{P}_{\text{rel}}$ be the involutions on $\mathcal{A}_i$ and $\mathcal{B}_i$ defined in Lemma 10. Since they are involutions, their eigenvalues are either $+1$ or $-1$. We write their eigenspace decompositions as

$$\mathcal{A}_i = \mathcal{A}_i^+ \oplus \mathcal{A}_i^- \qquad \text{and} \qquad \mathcal{B}_i = \mathcal{B}_i^+ \oplus \mathcal{B}_i^-,$$

where the superscripts $\pm$ denote the eigenspaces corresponding to eigenvalues $\pm 1$, respectively.

Recall from Lemma 10 that $(\mathcal{P}_{\text{pair}} \otimes \mathcal{P}_{\text{rel}})(\Phi_i) = \Phi_i$. Note that the operator $\mathcal{P}_{\text{pair}} \otimes \mathcal{P}_{\text{rel}}$ acts on a pure tensor $U \otimes S \in \mathcal{A}_i^{\sigma} \otimes \mathcal{B}_i^{\tau}$ (where $\sigma, \tau \in \{+, -\}$) by scalar multiplication with $\sigma \cdot \tau$. Since $\Phi_i$ is invariant (eigenvalue $+1$), it must lie in the subspace where this product is positive:

$$\Phi_i \in (\mathcal{A}_i^+ \otimes \mathcal{B}_i^+) \oplus (\mathcal{A}_i^- \otimes \mathcal{B}_i^-).$$

Consequently, $\Phi_i$ uniquely decomposes as $\Phi_i = \Phi_i^+ + \Phi_i^-$ with terms in these respective subspaces.

We now expand each $\Phi_i^\pm$ as a finite sum of pure tensors with factors in the corresponding eigenspaces. Since $V_{\text{ctx}}$ and $V_{\text{rel}}$ are finite-dimensional (in particular, $|E|, |R| < \infty$), the spaces $\mathcal{A}_i^\pm$ and $\mathcal{B}_i^\pm$ are finite-dimensional. Let $d_+ := \dim \mathcal{A}_i^+$ and choose a basis $\{U_{i,1}^+, \ldots, U_{i,d_+}^+\}$ of $\mathcal{A}_i^+$. Consider the linear map

$$T^+ : (\mathcal{B}_i^+)^{d_+} \to \mathcal{A}_i^+ \otimes \mathcal{B}_i^+, \qquad T^+(S_1, \ldots, S_{d_+}) := \sum_{k=1}^{d_+} U_{i,k}^+ \otimes S_k.$$

This map is surjective. Indeed, any element in the tensor product is a sum of pure tensors $U \otimes S$. Since $\{U_{i,k}^+\}$ is a basis, any $U \in \mathcal{A}_i^+$ can be written as $\sum_k c_k U_{i,k}^+$. Substituting this expansion into the pure tensors and regrouping terms by $U_{i,k}^+$ shows that any element takes the form $\sum_k U_{i,k}^+ \otimes S_k'$ for some $S_k' \in \mathcal{B}_i^+$.

Since the dimensions of the domain and codomain coincide:

$$\dim(\mathcal{B}_i^+)^{d_+} = d_+ \cdot \dim \mathcal{B}_i^+ = (\dim \mathcal{A}_i^+) \cdot (\dim \mathcal{B}_i^+) = \dim(\mathcal{A}_i^+ \otimes \mathcal{B}_i^+),$$

the surjective linear map $T^+$ is an isomorphism. Therefore, there exist unique maps $S_{i,1}^+, \ldots, S_{i,d_+}^+ \in \mathcal{B}_i^+$ such that

$$\Phi_i^+ = \sum_{k=1}^{d_+} U_{i,k}^+ \otimes S_{i,k}^+.$$

Set $I_i^+ := \{1, \ldots, d_+\}$. Analogously, we obtain unique maps $S_{i,1}^-, \ldots, S_{i,d_-}^- \in \mathcal{B}_i^-$ such that

$$\Phi_i^- = \sum_{k=1}^{d_-} U_{i,k}^- \otimes S_{i,k}^-.$$

Set $I_i^- := \{1, \ldots, d_-\}$.

Now, define the projected feature maps by evaluation on basis tensors:

$$\phi_i^\pm(h, r, t) := \Phi_i^\pm(e_{(h,t)} \otimes e_r), \qquad u_{i,k}^\pm(h, t) := U_{i,k}^\pm(e_{(h,t)}), \qquad v_{i,k}^\pm(r) := S_{i,k}^\pm(e_r).$$

Under the identification of Lemma 9, each pure tensor $U_{i,k}^\pm \otimes S_{i,k}^\pm$ acts by $(x \otimes y) \mapsto U_{i,k}^\pm(x) \otimes S_{i,k}^\pm(y)$, hence

$$\phi_i^\pm(h, r, t) = \sum_{k \in I_i^\pm} u_{i,k}^\pm(h, t) \otimes v_{i,k}^\pm(r),$$

and $\phi_i = \phi_i^+ + \phi_i^-$ since $\Phi_i = \Phi_i^+ + \Phi_i^-$. Finally, since $U_{i,k}^\pm \in \mathcal{A}_i^\pm$ means $U_{i,k}^\pm \circ P_{\text{pair}} = \pm U_{i,k}^\pm$, evaluating at $e_{(h,t)}$ gives $u_{i,k}^\pm(t, h) = \pm u_{i,k}^\pm(h, t)$. Likewise $S_{i,k}^\pm \in \mathcal{B}_i^\pm$ means $S_{i,k}^\pm \circ P_{\text{rel}} = \pm S_{i,k}^\pm$, and evaluating at $e_r$ gives $v_{i,k}^\pm(r^\smile) = \pm v_{i,k}^\pm(r)$. Summing over $i$ yields the global decomposition. $\qquad\square$

## E. Proof of Lemma 2

In this section, we provide the proof of Lemma 2.

**Lemma 2** (Kernel Stability Yields Bilinearity). *Assume Assumption 1 and the symmetry of conjunction (Def. 4(ii)). Then, there exists a unique symmetric bilinear operator $\tilde{F} : W^\wedge \times W^\wedge \to W^\wedge$ such that $\tilde{F}(\phi_p, \phi_q) = \phi_{p \wedge q}$ for all $p, q \in Q^\wedge$.*

*Proof.* Fix $q \in Q^\wedge$. Let $V^\wedge$ be the free real vector space with basis $\{e_p : p \in Q^\wedge\}$, and let $\Phi : V^\wedge \to W^\wedge$ be the linear map defined on basis vectors by $\Phi(e_p) = \phi_p$. Define a linear map $P_{\wedge,q} : V^\wedge \to V^\wedge$ by its action on basis:

$$P_{\wedge,q}(e_p) := e_{p \wedge q} \qquad (p \in Q^\wedge).$$

First, we claim that $P_{\wedge,q}$ induces a well-defined linear operator $L_\wedge(q) \in \text{End}(W^\wedge)$ satisfying

$$L_\wedge(q)\, \phi_p = \phi_{p \wedge q} \qquad \forall p \in Q^\wedge.$$

For any $w \in W^\wedge$, since $W^\wedge = \mathrm{span}\{\phi_p : p \in Q^\wedge\}$, we may choose $x \in V^\wedge$ with $\Phi(x) = w$, and define

$$L_\wedge(q)\, w := \Phi\big(P_{\wedge,q}x\big).$$

It remains to check that this does not depend on the choice of $x$. If $x'$ is another preimage of $w$, then $x - x' \in \ker \Phi$. By Assumption 1, we have $P_{\wedge,q}(\ker \Phi) \subseteq \ker \Phi$, hence $\Phi(P_{\wedge,q}(x - x')) = 0$, i.e., $\Phi(P_{\wedge,q}x) = \Phi(P_{\wedge,q}x')$. Thus $L_\wedge(q)$ is well-defined. Linearity follows from linearity of $\Phi$ and $P_{\wedge,q}$. Applying the definition to $w = \phi_p = \Phi(e_p)$ gives

$$L_\wedge(q)\, \phi_p = \Phi\big(P_{\wedge,q}e_p\big) = \Phi(e_{p \wedge q}) = \phi_{p \wedge q},$$

as claimed.

We now construct the map $\tilde{F} : W^\wedge \times W^\wedge \to W^\wedge$. Since $\{\phi_p : p \in Q^\wedge\}$ spans $W^\wedge$, any vector $v \in W^\wedge$ can be decomposed as a finite linear combination $v = \sum_i c_i \phi_{q_i}$. For any $u \in W^\wedge$, we define $\tilde{F}$ linear in the second argument by using the operators $L_\wedge(q_i)$:

$$\tilde{F}(u, v) := \sum_i c_i L_\wedge(q_i)\, u.$$

To ensure the well-definedness of $\tilde{F}(u, v)$, we must check that this definition is independent of the decomposition of $v$. Suppose $\sum_i c_i \phi_{q_i} = 0$, i.e., the vector $y = \sum_i c_i e_{q_i}$ lies in $\ker \Phi$. We check the value of the map for any basis vector $u = \phi_p$ ($p \in Q^\wedge$):

$$\sum_i c_i L_\wedge(q_i)\, \phi_p = \sum_i c_i \phi_{p \wedge q_i}.$$

By commutativity of conjunction on realized inputs (equivalently, by the assumed symmetry of $F$), we have $\phi_{p \wedge q_i} = \phi_{q_i \wedge p}$ for all $i$. Thus,

$$\sum_i c_i \phi_{p \wedge q_i} = \sum_i c_i \phi_{q_i \wedge p} = \Phi\left(\sum_i c_i e_{q_i \wedge p}\right) = \Phi\big(P_{\wedge,p}(y)\big).$$

Since $y \in \ker \Phi$, by Assumption 1, we have $P_{\wedge,p}(y) \in \ker \Phi$, so $\Phi(P_{\wedge,p}(y)) = 0$. Thus, $\sum_i c_i L_\wedge(q_i)\phi_p = 0$ for all $p \in Q^\wedge$. Since $\{\phi_p : p \in Q^\wedge\}$ spans $W^\wedge$, this implies $\sum_i c_i L_\wedge(q_i)u = 0$ for all $u \in W^\wedge$. Therefore, if $v = \sum_i c_i \phi_{q_i} = \sum_j d_j \phi_{r_j}$ are two decompositions, then $\sum_i c_i \phi_{q_i} - \sum_j d_j \phi_{r_j} = 0$ implies

$$\sum_i c_i L_\wedge(q_i)u = \sum_j d_j L_\wedge(r_j)u \qquad \forall\, u \in W^\wedge,$$

so $\tilde{F}(u, v)$ is independent of the chosen decomposition of $v$.

By construction, $\tilde{F}$ is linear in the second argument. Also, since each $L_\wedge(q_i)$ is linear, $\tilde{F}$ is linear in the first argument. Finally, for realized inputs, $\tilde{F}(\phi_p, \phi_q) = L_\wedge(q)\phi_p = \phi_{p \wedge q}$. Thus, $\tilde{F}$ is the unique bilinear extension. $\qquad\square$

## F. Robustness to Approximate Negation

In Theorem 3, negation equivariance was assumed exactly, i.e., $\phi_{\neg p} = -\phi_p$. In this section, we demonstrate a relaxed version showing that the obstruction does not disappear under a small violation of exact negation equivariance.

Let $u := \phi_p$ and suppose that negation is $\varepsilon$-approximately represented by a sign flip:

$$\phi_{\neg p} = -u + \delta, \qquad \|\delta\| \leq \varepsilon \|u\|.$$

Assume that $\tilde{F} : W^\wedge \times W^\wedge \to W^\wedge$ is the bilinear conjunction operator from Lemma 2, and let

$$M := \|\tilde{F}\|_{\mathrm{op}} = \sup_{\|a\|=\|b\|=1} \|\tilde{F}(a, b)\|.$$

For simplicity, assume $\|u\| = 1$. By idempotence, $\tilde{F}(u, u) = u$ and

$$\tilde{F}(\phi_{\neg p}, \phi_{\neg p}) = \phi_{\neg p} = -u + \delta.$$

On the other hand, by bilinearity,

$$\tilde{F}(\phi_{\neg p}, \phi_{\neg p}) = \tilde{F}(-u + \delta, -u + \delta)$$
$$= \tilde{F}(u, u) - \tilde{F}(u, \delta) - \tilde{F}(\delta, u) + \tilde{F}(\delta, \delta)$$
$$= u - \tilde{F}(u, \delta) - \tilde{F}(\delta, u) + \tilde{F}(\delta, \delta).$$

Equating the two expressions gives

$$2u = \delta + \tilde{F}(u, \delta) + \tilde{F}(\delta, u) - \tilde{F}(\delta, \delta).$$

Taking norms and using the definition of $M$ yields

$$2 \le \|\delta\| + \|\tilde{F}(u, \delta)\| + \|\tilde{F}(\delta, u)\| + \|\tilde{F}(\delta, \delta)\|$$
$$\le \varepsilon + 2M\varepsilon + M\varepsilon^2 = \varepsilon(1 + 2M + M\varepsilon).$$

Thus, approximate negation can coexist with bilinear idempotent conjunction only if

$$2 \le \varepsilon(1 + 2M + M\varepsilon).$$

In particular, when the bilinear conjunction is normalized so that $M = 1$, this requires

$$\varepsilon \ge \frac{\sqrt{17} - 3}{2} \approx 0.56.$$

Therefore, the exact collapse in Theorem 3 is robust in the sense that small perturbations of negation equivariance are still incompatible with a bounded bilinear idempotent conjunction.

