# OpenReview forum: "Dynamics Reveals Structure: Challenging the Linear Propagation Assumption"
_ICML.cc/2026/Conference — ICML 2026 spotlight_

### Official Review · Reviewer_MgGj · 2026-03-11

**Soundness:** 2
**Presentation:** 3
**Significance:** 3
**Originality:** 3
**Overall Recommendation:** 4
**Confidence:** 1

**Summary:**

This paper investigates the geometric limits of LPA under first-order parameter updates and proposes a theoretical framework based on relational algebra and gradient linearized features. Within this framework, the authors derive necessary structural conditions on negation/converse. Specifically, to guarantee direction-agnostic first-order propagation, negation requires a tensor factorization that separates entity-pair context from relation content (Theorem 1). Converse further requires a decomposition into symmetric and antisymmetric components, constraining how argument order is represented (Theorem 2). Moreover, the paper identifies a fundamental obstruction when extending this systematicity to composition (Theorem 3).

**Compliance With Llm Reviewing Policy:**

Affirmed.

**Final Justification:**

This rebuttal addresses my primary concern about the lack of empirical support for the theoretical claims.

**Key Questions For Authors:**

1. This paper adopts relation algebra and study three core operations on relations. These results suggest that failures in knowledge editing, the reversal curse, and multi-hop reasoning may stem from common structural limitations inherent to the LPA. However, the paper currently lacks empirical support for these claims, making it difficult to assess whether the theory faithfully captures behavior observed in practice. A toy model or controlled synthetic experiment could be a useful first step to support the theory.

**Limitations:**

I suggest that the authors explore the limitations of the theoretical conclusions in practical applications. Having not empirically verified the theoretical results, I do not know the reliability of the theoretical results in practical applications.

**Strengths And Weaknesses:**

Strengths

1. This study empirically presents that the current LLMs violate a basic requirement for negation consistency under LPA.

2. By introducing relation algebra and linearized Features to give a central definition: Systematic Linear Propagation.

3. The analysis reveals several necessary conditions on the required geometry of first-order updates.

Weaknesses

1. The theoretical results are highly dependent on SLP and Assumption 1 obtained from relation algebra and linearization features, but the SLP definition is strong. For example, SLP requires that the logical constraints hold for all  $\Delta\theta\in W$ in the first-order regime, as well as in practical applications where $\phi$ is a learned low-dimensional embedding, Assumption 1 usually does not hold.

2. The theoretical results lack empirical support.

---

> ### Author Rebuttal · Authors · 2026-03-31
>
> We thank Reviewer MgGj for the time and effort in reviewing our work. We are glad that the reviewer has recognized our contribution to analyzing logical consistency under knowledge updates. We respond to each concern as follows:
>
> ### **W1:** SLP is strong & Assumption 1 may not hold
>
> **1. Justification for the SLP**
> SLP is deliberately strong. As noted in Sec. 2.4, we formalize it as a direction-agnostic requirement to test whether logical coherence is an intrinsic property of the local geometry. However, its strictness is intended to precisely analyze and expose the baseline structural limits of current local first-order update paradigms.
>
> If SLP fails, maintaining logical coherence is no longer automatic from the local geometry. It instead depends on selecting update directions that satisfy additional preservation and propagation constraints. True, existing local editing (and unlearning) methods are carefully engineered to mitigate interference. Yet, selecting update directions using only limited auxiliary constraints does not by itself guarantee the preservation of all facts one wishes to preserve, or the correct changes on logical consequences. To theoretically address this, SLP isolates the regime in which these preservation and propagation requirements are handled automatically by the local geometry.
>
> **2. Justification for Assumption 1**
> Regarding Assumption 1, we would like to clarify that the feature $\phi_p$ is not a learned low-dimensional embedding; it represents the target-logit gradient in the parameter space ($\phi_p = \nabla_\theta s(p)$). Thus, its dimensionality can be as large as the model's parameter count.
>
> Assumption 1 requires that if a linear combination vanishes, then its conjunction with any fixed context must also vanish. One can certainly argue that this assumption might not hold perfectly in practical networks. However, rejecting it implies that conjunction can no longer be systematically treated as a well-defined linear operator on these gradient features. In other words, the rejection makes it difficult to maintain the premise that compositional reasoning can systematically emerge through linear propagation alone. Consequently, the potential empirical failure of Assumption 1 does not invalidate our framework. Instead, it highlights why first-order updates struggle to support multi-hop reasoning consistently.
>
> ### W2 & Q1: Lack of empirical support & Any controlled experiment?
>
> We appreciate the reviewer's thoughtful suggestion for a controlled experiment. We agree that empirically connecting our theoretical analysis to the failures in knowledge editing, the reversal curse, and multi-hop reasoning can be strengthened. While a toy setup could illustrate the theoretically proved mechanisms, we instead provide new evidence in a real LLM setting to support the key premises of our theory.
>
> First, we provide a new experiment supporting that the linear regime is practically relevant for local LLM updates. Inspired by common local editing setups that target mid-early MLP layers (Meng et al., 2022a), we experimented on Qwen3-4B-Instruct using 100 TREx instances at layer 10. We updated the parameters along the target-logit gradient direction ($g/\|g\|$) with varying step sizes ($\eta$) and compared the actual logit change against the expected first-order linear approximation ($\eta \cdot \|g\|$). The linear regression was performed across the instances to find the slope and $R^2$ values.
>
> | Step Size ($\eta$) | Slope (Actual vs Expected) | $R^2$ |
> | --- | --- | --- |
> | 1e-05 | 0.97 | >0.99 |
> | 1e-04 | 1.00 | >0.99 |
> | 1e-03 | 0.99 | >0.99 |
> | 1e-02 | 0.89 | 0.98 |
> | 1e-01 | 0.21 | 0.19 |
>
> The result suggests that the model behavior remains well-approximated by first-order geometry ($R^2 \approx 1.0$) over practically relevant step sizes (up to $10^{-2}$). This supports the practical relevance of our first-order framework. Within this same regime, our theory identifies two complementary limitations: Thm. 3 gives a structural obstruction to conjunction/composition, while Fig. 2 and 4 show that a necessary geometric condition for negation-consistent propagation is violated in practical LLMs. They are consistent with recent empirical reports where editing methods struggle to propagate edits to logical consequences (Cohen et al., 2024; Liu et al., 2025). Overall, the theory and the new empirical evidence support a common geometric interpretation of these phenomena: such failures plausibly arise when local first-order updates are expected to satisfy more logical systematicity than their geometry can guarantee.
>
> We thank the reviewer once again for your time and effort.

---

> > ### Author Rebuttal · Reviewer_MgGj · 2026-04-03
> >
> > Thank you for providing the empirical evidence. This addresses my main concern regarding the lack of experimental support for the theory. In light of this clarification, I am updating my score to Weak accept.

---

> > > ### Author Response · Authors · 2026-04-05
> > >
> > > We greatly appreciate the reviewer's willingness to re-evaluate our work in light of the additional empirical evidence. Thank you for the constructive feedback that motivated these improvements.

---

### Official Review · Reviewer_apAP · 2026-03-12

**Soundness:** 4
**Presentation:** 3
**Significance:** 3
**Originality:** 3
**Overall Recommendation:** 4
**Confidence:** 3

**Summary:**

This paper explores what the authors call the "Linear Propagation Assumption" (LPA). Simply put, LPA is the assumption that first-order parameter updates preserve logical coherence about the facts being updated as well as related beliefs not explicitly updated. Prior research has shown that LLMs generally do not maintain such logical coherence. Thus, the authors investigate what constraints must be satisfied to allow such coherence to be maintained, which they formalize as "Systematic Linear Propagation" (SLP). The authors interpret the relations being updated in terms of relational algebra and use this formalization to prove that the negation and converse cases are theoretically satisfiable by linear updates to the LLM but that conjunction requires a constraint incompatible with negation. Thus, the overall conclusion is that linear updates to the model fundamentally constrain logical coherence of the updates, suggesting that alternative update geometries are necessary to guarantee logical coherence during updates.

**Compliance With Llm Reviewing Policy:**

Affirmed.

**Final Justification:**

The rebuttal addressed my questions and I maintain my original score.

**Key Questions For Authors:**

1. Perhaps I missed this, but I did not find anything where this analysis was specific to LLMs. Does any proof or setting use assumptions specific to language models?
2. Although you prove that the negation and conjunction requirements are incompatible, it's not clear to me the implications of that incompatibility. Is it possible to formalize something like error bounds?
3. Under the requirements imposed by your formalization, how might you extend this analysis for multi-step non-local optimization?

**Limitations:**

yes

**Strengths And Weaknesses:**

Strengths:
1. The paper analyzes LLM updates under a creative regime, applying the relational algebra to knowledge updates.
2. The claims of the paper are technically sound, supported either by prior results or derived within the setup of the paper.
3. The topic of the paper is especially relevant in current times where commonly used tools are being replaced by pure-AI alternatives, increasing the risk of misinformation due to slight differences in how knowledge is queried.

Weaknesses:
1. There paper is rich in notation and terminology which may not be familiar to NLP/ML audiences compared to database audiences. For example, the notation for closure is first used over 3 pages after it is explained, at which point the actual composition is highly important for understanding the subsequent proofs.
2. It seems that the conclusions of the analysis in this paper only apply to individual updates, but it's unclear if these results apply when linear updates are iteratively applied over multiple optimization steps throughout the model.
3. (Minor) The authors suggest properties that optimization methods may need to satisfy the logical coherence constraints but do not provide concrete guidance to analyze any such method beyond that they must enforce the theorized structures. I suspect that this analysis framework will be difficult to apply in practice to prove theoretical coherence.

---

> ### Author Rebuttal · Authors · 2026-03-31
>
> We thank Reviewer apAP for the time and effort in reviewing our work. We are glad that the reviewer recognized the creativity, technical soundness, and timeliness of our contribution. We address the questions and concerns below.
>
> ### W1: Notation & terminology
>
> We thank the reviewer for pointing this out. We will improve accessibility for general NLP/ML audiences by reintroducing the key notation at the point where it becomes important for the later proofs. In particular, after Def. 4, we will explicitly remind the reader as follows:
>
> > By the closure $Q^\wedge$ of $Q$ under negation and conjunction, we denote the smallest set of queries containing $Q$ and closed under negation and conjunction.
>
> ### W2 & Q3: Extension to multi-step & non-local optimization
> We appreciate the insightful question.
> Our theorems are local by design: they characterize what first-order geometry can guarantee at a reference point $\theta_0$. For multi-step optimization, the natural extension is the NTK / lazy-training regime: when the tangent features remain approximately constant during training, gradient descent is governed by the same first-order geometry at every step (Jacot et al., 2018; Lee et al., 2019), and finite-horizon bounds between the nonlinear and linearized trajectories can be established (Chizat et al., 2019). In that regime, the same first-order obstruction applies stepwise along the trajectory, rather than only to a single isolated update.
> However, for genuinely long, non-local optimization where higher-order effects dominate, the obstruction identified by Thm. 3 need not directly apply, as we discussed as a potential future direction in Sec. 6.
> We will add this discussion to the revised manuscript.
>
> ### W3: Practical application of the framework
>
> We believe that our framework can be highly practical as a diagnostic tool. Concretely, one can test: (i) whether a model stays in a linearized regime by comparing actual score changes to $\langle \phi_q, \Delta\theta \rangle$; (ii) whether features admit approximate context-relation factorization by measuring how much variance in the feature tensor $T_{(h,t),r}$ is explained by a separable model $\sum_k u_k(h,t) \otimes v_k(r)$; and (iii) probing whether compound-query features are well approximated by a bilinear map. We will include a discussion on these diagnostics to guide future method design.
>
> ### Q1: Assumptions specific to language models?
>
> We did not use any assumptions specific to Large Language Models or their specific architectures (like Transformers). Our analysis universally applies to any connectionist network updated via first-order gradient signals. The LLM framing is deliberate because the motivating failure modes (ripple effects, reversal, and multi-hop propagation) are especially well documented there. Our formulation is exact in the NTK regime, and LLMs are a practically relevant case of overparameterized models where local linearization is plausible. Therefore, our framework can be naturally extended to analyze logical propagation in other domains, such as vision models.
>
> ### Q2: Implications of incompatibility and error bounds
>
> To formalize the implications of the incompatibility, we can introduce an error bound by defining an $\varepsilon$-approximate negation.
>
> Let $\phi_{\neg p} = -\phi_p + \delta$ with $\|\delta\| \leq \varepsilon\|\phi_p\|$.
> With the exact assumption ($\varepsilon=0$), combining bilinearity $\tilde{F}(-\phi_p, -\phi_p) = \tilde{F}(\phi_p, \phi_p)$ with idempotence yields $\phi_p = -\phi_p$, hence a collapse ($\phi_p = 0$).
> With the relaxation, substituting $\phi_{\neg p} = -\phi_p + \delta$ into idempotence and expanding bilinearly leaves:
> $2\phi_p = \delta + \tilde{F}(\phi_p, \delta) + \tilde{F}(\delta, \phi_p) - \tilde{F}(\delta, \delta)$
>
> Taking norms (writing $M = \|\tilde{F}\|_{\mathrm{op}}$ and assuming unit-norm features) yields the bound:
> $2 \leq \varepsilon(1 + 2M + M\varepsilon)$
>
> This bound shows that near-exact negation cannot coexist with linear conjunction: even in the most favorable case ($M = 1$), one needs $\varepsilon \ge (\sqrt{17}-3)/2 \approx 0.56$.
>
> This demonstrates that the structural collapse of linear conjunction does not vanish gracefully even when the strict symmetric assumptions of logical equivariance are relaxed. Our experiments in Fig. 2 and 4 report a positive cosine between $\phi_p$ and $\phi_{\neg p}$. Therefore, any decomposition $\phi_{\neg p} = -\phi_p + \delta$ must satisfy $\|\delta\| > \|\phi_p\|$, i.e., $\varepsilon > 1$. Indeed, $\langle \phi_p, \phi_{\neg p} \rangle > 0$ gives $\langle \phi_p, -\phi_p + \delta \rangle > 0$, hence $\|\delta\| > \|\phi_p\|$. Thus, current LLMs lie far outside even a weak approximate-negation regime, which is consistent with the observed difficulty of negation propagation in editing (Liu et al, 2025).
>
> We thank the reviewer once again for your time and effort.
>
> (Chizat et al., 2019): On Lazy Training in Differentiable Programming

---

> > ### Author Rebuttal · Reviewer_apAP · 2026-04-04
> >
> > Thank you for your detailed response. This is relatively in line with my assumptions while reviewing, hence, I will maintain my positive score. Nice work!

---

> > > ### Author Response · Authors · 2026-04-05
> > >
> > > We are grateful for the reviewer's thorough engagement with our work and the constructive questions that helped us strengthen the paper. Thank you for acknowledging our responses and for the positive assessment.

---

### Official Review · Reviewer_Qvqe · 2026-03-12

**Soundness:** 3
**Presentation:** 3
**Significance:** 3
**Originality:** 4
**Overall Recommendation:** 4
**Confidence:** 3

**Summary:**

This paper investigates the logical systematicity under the linear propagation assumption, which leverages the first-order Taylor expansion. Logical systematicity of negation, converse, and conjunction (multi-hop) is discussed. Interesting, the co-existence of linear feature space (which is the gradient of the score) and requirement of logical first principles lead to the failure of the linear propagation assumption. Suggesting the pathway for future research.

**Compliance With Llm Reviewing Policy:**

Affirmed.

**Key Questions For Authors:**

1. whether the formal assumptions are too strong for practical model editing? Would there be any empirical evidences?
2. one of the key element is that the negation is achieved by sign flipping. However, this is not supported by evidence in Figure 2. Does this fact weaken the practical significance of this discussion?

**Limitations:**

Yes.

**Strengths And Weaknesses:**

Strength:
- This paper reveals the key deficiency of one fundamental assumption LPA in the field of knowledge editing by undeniable first principles rooted in logical algebras and linear geometrical spaces. Revealing the compiling fact that such geometric assumption failed to capture the systematicity of the logical structure.
- The analysis is convincing and applies not only to LLMs but all connectionism networks.
- The implication of this paper is also meaningful. Even though LPA will almost surely be a minimal way to attack related problems. Knowing its gap is beneficial.

Weakness:
- Some notations are not clearly stated properly, such as the O sum and O times, which is not commonly defined for readers in general background. Adding a paragraph describing notations somewhere are welcomed.

---

> ### Author Rebuttal · Authors · 2026-03-31
>
> We thank Reviewer Qvqe for the time and effort in reviewing our work. We are glad that the reviewer has recognized our work's originality, significance, and broad applicability to any connectionist network that relies on first-order parameter updates. We respond to each weakness and question as follows:
>
> ### W1: Not clearly stated notations
>
> We appreciate the thoughtful feedback. To improve accessibility, we will explain these concepts in the App. B primer, and add the following explanation after Thm. 1:
>
> > We use $\bigoplus_i W_i$ for a direct-sum decomposition: every feature vector has one independent component in each block $W_i$, and $C \otimes R$ for a tensor product: a context/entity-pair factor in $C$ bound to a relation factor in $R$.
>
> ### Q2: Sign flipping for negation is not supported by Fig. 2?
>
> The result in Fig. 2 **strengthens** our claim by falsifying a necessary condition derived from our theoretical framework. Our construction states that a necessary condition for SLP is that negation must be realized in the tangent manifold via a sign flip. Fig. 2 demonstrates this necessary condition is severely violated in practical LLMs. Because this condition is empirically violated, practical models cannot satisfy the negation requirement of SLP. This geometric mismatch offers a principled explanation for why editing methods struggle with negation propagation (Liu et al., 2025). We will make this point explicit in the revision.
>
> ### Q1: Are formal assumptions too strong? Empirical evidence?
>
> We would like to justify that our setups carry practical relevance, both logically and empirically:
>
> - **Justification of the linearized regime:** LPA is an implicit premise in most gradient-based update methodologies, as they rely on first-order information to find update directions. For example, prominent locate-and-edit methods such as ROME and MEMIT adopt a local linear-memory view of the edited MLP layer (Meng et al., 2022a;b). Therefore, analyzing logical propagation under a linearized regime reflects current practices.
>
>     To support the practical relevance of the linearized regime, we present an additional experiment. Following common local editing setups that target mid-early MLP layers, we experimented on Qwen3-4B-Instruct using 100 TREx instances at layer 10. We updated parameters along the target-logit gradient direction ($g/\|g\|$) with varying step sizes ($\eta$). Comparing the actual logit change against the expected first-order inner product ($\eta \cdot \|g\|$), the actual updates are well-predicted by the linear approximation before reaching very large step sizes ($\ge 10^{-1}$):
>
>     | Step Size ($\eta$) | Slope (Actual vs Measured) | $R^2$ |
>     | --- | --- | --- |
>     | 1e-05 | 0.97 | >0.99 |
>     | 1e-04 | 1.00 | >0.99 |
>     | 1e-03 | 0.99 | >0.99 |
>     | 1e-02 | 0.89 | 0.98 |
>     | 1e-01 | 0.21 | 0.19 |
> - **Justification of the SLP:** SLP is deliberately strong. As noted in Sec. 2.4, we formalize it as a direction-agnostic requirement to test whether logical coherence is an intrinsic property of the local geometry. Its strictness is intended to analyze and expose the baseline structural limits of current local first-order update paradigms. If it fails, maintaining logical coherence is no longer automatic from the local geometry; instead, it depends on selecting update directions that satisfy additional preservation and propagation constraints, which does not by itself guarantee preservation of all facts one wishes to preserve or propagate. To theoretically address this, SLP isolates the regime in which these requirements are handled automatically by the local geometry.
>
> - **Justification of Assumption 1:** Assumption 1 requires that if a linear combination vanishes, then its conjunction with any fixed context must also vanish. One can argue that this assumption may not hold perfectly in practice. However, rejecting it implies that conjunction can no longer be systematically treated as a well-defined linear operator on these gradient features, making it difficult to maintain the premise that compositional reasoning can emerge through linear propagation alone. Thus, the potential empirical failure of Assumption 1 does not invalidate our framework.
> - **The Collapse Persists Under Relaxed Conditions:** We can also relax exact negation to $\varepsilon$-approximate negation. If $\phi_{\neg p} = -\phi_p + \delta$ with $\|\delta\| \le \varepsilon\|\phi_p\|$, then substituting into idempotence and expanding bilinearly gives: $2\phi_p = \delta + \tilde{F}(\phi_p, \delta) + \tilde{F}(\delta, \phi_p) - \tilde{F}(\delta, \delta)$. Taking norms (writing $M = \|\tilde{F}\|_{\mathrm{op}}$ and assuming unit-norm features): $2 \le \varepsilon(1 + 2M + M\varepsilon)$. Even in the most favorable case ($M = 1$), this forces $\varepsilon \ge (\sqrt{17}-3)/2 \approx 0.56$. So the collapse does not disappear under a small perturbation of exact negation.
>
> We thank the reviewer again for your time and effort.

---

> > ### Author Rebuttal · Reviewer_Qvqe · 2026-04-04
> >
> > The authors have provided in-depth explanations. I would like to keep my positive support.

---

> > > ### Author Response · Authors · 2026-04-05
> > >
> > > We sincerely thank the reviewer for the thoughtful evaluation and for acknowledging that our responses have fully addressed the concerns. We would be grateful if the reviewer might consider reflecting these improvements in the overall score. Thank you again for your support and valuable feedback.

---

### Official Review · Reviewer_VLWc · 2026-03-13

**Soundness:** 4
**Presentation:** 3
**Significance:** 3
**Originality:** 4
**Overall Recommendation:** 5
**Confidence:** 3

**Summary:**

The manuscript studies the limitations of neural network training via
gradient descent in terms of consistency with respect to logical
operations. It highlights how standard gradient updates fail to
guarantee consistency and provides a principled framework to study
conditions guaranteeing consistency wrt various relations.

**Compliance With Llm Reviewing Policy:**

Affirmed.

**Final Justification:**

This is a solid contribution on the learnability problems of gradient-based approaches, providing a fresh perspective on some difficulties observed in learning certain patterns of reasoning.

**Key Questions For Authors:**

none

**Limitations:**

yes

**Strengths And Weaknesses:**

The paper sheds new light on the reasons for well-known failure modes
in standard neural models like LLMs, like management of negation and
multi-hop reasoning.

The problem is formalized in a rigorous and actionable way using
relation algebra, which allows to identify geometric constraints
needed to achieve logical consistency and incompatibility between
different relations when it comes to guaranteeing consistency with
gradient updates.

---

> ### Author Rebuttal · Authors · 2026-03-31
>
> We thank Reviewer VLWc for the time and effort in reviewing our work. We are glad that the reviewer has recognized the originality of our work, its contribution to shedding light on commonly observed failure cases in neural models, and the rigor of our formalization.
>
> We would like to briefly note two additions that further strengthen the paper and will be incorporated in the revision.
>
> - First, we extend our theoretical framework to include an error-bound analysis using an $\varepsilon$-approximate negation. This quantitative analysis shows that the obstruction to linear conjunction persists even under relaxed negation, unless the geometry departs substantially from exact sign-flip behavior.
>
> - Second, we add an empirical experiment supporting the practical relevance of the first-order linearized regime in local LLM editing. Adopting common local editing practices (i.e., targeting specific MLP layers) and evaluating the empirical scaling of target-logit gradient updates, the results show that actual logit changes are closely tracked by a first-order linear approximation for step sizes smaller than $10^{-2}$. This suggests that local LLM editing is well approximated by a linearized geometry over the typical range of step sizes, thereby supporting the practical relevance of our first-order analysis.
>
> We thank the reviewer once again for your time and effort in supporting and evaluating our research.

---

> > ### Author Rebuttal · Reviewer_VLWc · 2026-04-03
> >
> > Thanks a lot for the additional insights, that contribute to strengthen an already solid contribution.

---

> > > ### Author Response · Authors · 2026-04-05
> > >
> > > We sincerely appreciate the reviewer's continued support and kind words on our contribution. Thank you for taking the time to engage with our rebuttal and for the encouraging feedback throughout the review process.

---

### Decision · Program_Chairs · 2026-04-30

**Decision:**

Accept (spotlight)

**Comment:**

Reviewers agree this paper makes creative use of relational algebra to offer new insights into the cause of several well-known failure modes in LLMs.